



# Upscaling land-use effects on water partitioning and water ages using tracer-aided ecohydrological models

Aaron A. Smith[1], Doerthe Tetzlaff[1,2,3], Lukas Kleine[1,2], Marco Maneta[4], Chris Soulsby[3]

[1]IGB Leibniz Institute of Freshwater Ecology and Inland Fisheries Berlin, Berlin, Germany
[2]Humboldt University Berlin, Berlin, Germany
[3]Northern Rivers Institute, School of Geosciences, University of Aberdeen, UK
[4]School of Geosciences, University of Montana, USA

*Correspondence to*: Aaron A. Smith (smith@igb-berlin.de)

**Abstract.** Quantifying how vegetation mediates water partitioning at different spatial and temporal scales in complex,
managed catchments is fundamental for long-term sustainable land and water management. Estimations from ecohydrological models conceptualizing how vegetation regulates the inter-relationships between catchment water storage dynamics, evapotranspiration losses, and recharge/runoff fluxes are needed to assess water availability for a range of ecosystem services; and evaluate how these might change under increasing extreme events, such as droughts. Currently, the feedback mechanisms between water and mosaics of different vegetation/land cover are not well understood across spatial
scales and the effects of scale on the skill of ecohydrological models needs to be clarified. We used the tracer-aided ecohydrological model EcH$_2$O-iso in an intensively monitored 66 km$^2$ mixed land-use catchment in NE Germany to quantify water flux-storage-age interactions at four model-grid resolutions (250, 500, 750, and 1000m). This used a fusion of field (including precipitation, soil water, groundwater, and stream isotopes) and remote sensed data in the calibration. Multi-criteria calibration across the catchment at each resolution revealed some differences in the estimation of fluxes, storages,
and water ages. Larger grid-resolutions were unable to replicate observed streamflow and distributed isotope dynamics in the way smaller pixels could. However, using isotope data in the calibration still helped in constraining the estimation of fluxes, storage and water ages at coarser resolutions. Despite using the same data and parameterisation for calibration at different grid resolutions, the modelled proportion of fluxes differed slightly at each resolution, with coarse models simulating higher evapotranspiration, lower relative transpiration, increased overland flow, and slower groundwater movement. Although the
coarser resolutions also revealed higher uncertainty and lower overall model performance, the overall results were broadly consistent. The study shows that tracers provide effective calibration constraints on larger resolution ecohydrological modelling and help understand the influence of grid-resolution on the simulation of vegetation-soil interactions. This is essential in interpreting associated uncertainty in estimating land-use influence on large-scale "blue" (ground and surface water) and "green" (vegetation and evaporated water) fluxes, particularly for future environmental change.



## 1 Introduction

Climate projections indicate increases in temperatures and extreme drought frequency in many areas, with expected decreases in summer baseflow (Papadimitriou et al., 2016), reduced summer soil water storage (Grillakis, 2019), which in turn limit evapotranspiration (Jung et al., 2010). Under climatic change, there are concerns that long-term partitioning of blue (groundwater and stream water) and green (evapotranspiration, ET) water fluxes may be adversely affected by land

management for biomass production (i.e. agriculture and forestry) (Falkenmark and Rockström, 2006). However, there is a limited evidence base to project the likely relative effects of climate and land-use change on blue and green water fluxes (Orth and Destouni, 2018), and the associate predictive uncertainties (Mao et al., 2015). In regions susceptible to climatic extremes, there is a need to better quantify these fluxes to underpin sustainable long-term water and land-use policies for anthropogenic (drinking water abstraction and irrigation) and natural (forest, wetland and in-stream) ecosystem services.

Ecohydrological modelling provides an approach to quantify blue and green water fluxes and associated storage dynamics and project future change. Ecohydrological models can bridge a gap between complex hydrological and ecological processes and capture their integrated effect in controlling water partitioning in the critical zone – the thin layer of the Earth encompassing the top of the vegetation canopy down to the bottom of the groundwater (Grant and Dietrich, 2017; Brewer et al., 2018). The feedback between ecology and hydrology is, however, strongly scale-dependent, with controls on interactions

vastly different across space and time (Fatichi et al., 2015). The interdependency of models on the temporal and spatial scales often confounds identifiability of hydrological processes due to emergent behaviour, non-linearity of parameter interactions, and aggregation effects at coarser resolutions when models are applied at larger scales (Wood et al., 1988; Blöschl and Sivapalan, 1995; Horritt and Bates, 2001; Samaniego et al., 2017). Many of the advancements in addressing difficulties in model scaling have focused on discharge (Samaniego et al., 2017) or soil moisture (Vereecken et al., 2008)

due to limited alternative data and their information as proxies for large scale water availability and atmospheric exchange. Nevertheless, the complexities of soil-vegetation interactions mandate further clarification of scaling effects and the resolution boundaries of fluxes across ecohydrological interfaces in the critical zone (Krause et al., 2017; Vereecken et al., 2019).

Water stable isotope tracers (deuterium and oxygen-18; $\delta^2$H and $\delta^{18}$O, respectively) have been used as tools across various

regions and spatio-temporal scales to improve estimates of ecohydrological partitioning (Kool et al., 2014; Coenders-Gerrits et al., 2014; Jasechko, 2016). The integration of isotopic tracers in hydrological models has been shown to be an effective way of constraining ecohydrological flux and storage dynamic estimates at both small (Ala-aho et al., 2017; Kuppel et al., 2018; Knighton et al., 2020) and large scales (Stadnyk and Holmes, 2020; Holmes et al., 2020). Integration of tracers into modelling frameworks also allows non-stationary estimation of flux and storage water age dynamics for various critical zone

compartments. As water age studies have historically focused on groundwater and stream water dating of blue fluxes, estimation of water ages in process-based, semi-distributed ecohydrological models to characterise the "hydro-demographics" of evaporation and transpiration is fundamental to a more comprehensive understanding of ecohydrological



systems and their sensitivity to change (Kuppel et al., 2020). However, to date, such models have been usually been applied in smaller data-rich experimental catchments (<10 km$^2$).

The main aim of this study was to explore changes in the skill of an ecohydrological model in capturing flux, storage, and mixing dynamics across spatial scales through application to a mesoscale (~100 km2) catchment. We used the tracer-aided ecohydrological model EcH$_2$O-iso which couples physically-based hydrological conceptualisation with dynamic feedback mechanisms across the soil-plant-atmosphere continuum (Maneta and Silverman, 2013; Kuppel et al., 2018). EcH$_2$O-iso was developed with the intent of using diverse data in multi-criteria calibration, as well as interfacing with large-scale climate

models (Maneta and Silverman, 2013), through a fusion of field and remote sensed data. We seek to achieve the main aim of the study through the application of EcH$_2$O-iso in a drought-sensitive, agriculturally-dominated mesoscale catchment in north-eastern Germany for an 11-year model simulation period using four spatial resolutions. The study addresses three main research questions.

- Can a tracer-aided ecohydrological model effectively constrain estimates of water storage-flux-age interactions at
different spatial resolutions in larger, mixed land-use catchments?

- How does upscaling affect the robustness and uncertainty of model results in terms of parameterisation, sensitivity and calibration?

- What are the primary limitations of large resolution modelling vis-à-vis the use of calibration through field data (especially soil and stream isotopes), remote sensing data (e.g. MODIS or reanalysis data), or data fusion of field
and remote sensed data?

Evaluating these questions across different spatial model resolutions sought to provide a more robust understanding of the spatial boundaries of ecohydrological exchange, partitioning, and uncertainty in models. This is a prerequisite to using such models in decision support to inform land and water management.

## 2 Study site and data

### 2.1 Climate and model forcing data

The 66 km$^2$ Demnitzer Millcreek Catchment (DMC), is a mesoscale catchment 55 km east of Berlin (52$^o$23'N, 14$^o$15'E), receiving 575 mm of precipitation annually. Cumulative annual precipitation varies (372 to 776 mm/year); with summer usually slightly wetter than winter due to convectional storms, but winter is dominated by more frequent frontal rain (DWD, 2020). Potential evapotranspiration (PET) is very high relative to the annual precipitation (> 600 mm/year) and is generally

only less than annual precipitation in very wet years (UFZ, 2020; Smith et al., 2020a). Long-term average air temperature and relative humidity are 10$^o$C and 78%, respectively (Smith et al., 2020a; Smith et al., 2020b; Kleine et al., 2020).

Five long-term Deutscher Wetterdienst (DWD) stations surrounding the catchment were used for long-term assessment (Table 1). As local measurements of incoming short- and long-wave radiation were unavailable these were derived from remotely sensed data using ERA5 (ERA5, 2020). The ERA5 radiation is consistent with measurements near the DMC



(Douinot et al., 2019) and has been successfully used for the estimation of ET, transpiration (Tr), and latent heat (LE) (Smith et al., 2020b).

## 2.2 Soils and vegetation

The DMC land cover is dominated by non-irrigated arable crops in the northern headwaters and managed forests in the south; there is a long history artificial drainage, especially in wetlands in the central catchment (Fig. 1a) (Gelbrecht et al.,
2005). The general land-use is broadly representative of other extensive lowland agricultural areas in the North European Plain (e.g. Böse and Brande, 2010). The catchment is a long-term study site, with more than 30 years of monitoring agricultural pollution (Gelbrecht et al., 1996; Gelbrecht et al., 2005) and more recent detailed monitoring of stream isotopes, soil moisture, and soil isotopes (Smith et al., 2020a; Smith et al., 2020b; Kleine et al., 2020).

The catchment is characterised by four major soil types, with silty brown earths in the northern and southern regions, and
sandy gleys, peats and podzols dominating more central and southern regions (Fig. 1b). Brown earths are the most extensive soils (Fig. 1b, Table 2), and are siltier as a result of ground moraine deposited during the Pleistocene glaciation. Peats and sandy gley soils fringe the stream through the wetlands in the centre of the catchment and along the western edge of the catchment, respectively. The mid-catchment further from the stream is dominated by podzols and more sandy glacial deposits (Smith et al., 2020a).
Vegetation is categorised into four major groups: croplands (arable), pasturelands, broadleaf forests, and conifer forests (Table 2, Fig. 1c). Croplands, primarily consisting of winter wheat, barley, and maize, occupy higher quality soils in the North (Kleine et al., 2020). Much of the pastureland is in peat fens that are poorly drained nutrient-rich soils unsuitable for crops, and are therefore used for livestock grazing (Fig. 1c). Broadleaved forests are small and generally in the south covering a limited area (Table 2). Conifer forests are the second most common cover, dominating the south and generally
overlapping with the podzolic soils.

## 2.3 Hydrology of the Demnitzer Millcreek Catchment

Discharge is measured at Demnitz Mill and the catchment outlet (Fig. 1 and Table 1). Streamflow is groundwater-dominated, which results in a highly seasonal flow regime dependent on groundwater levels (Smith et al., 2020a). The catchment is situated on a large regional groundwater system that feeds the Spree River (Nützmann et al., 2014); however, regional
groundwater-surface interactions only impact the catchment near the southerly outlet (Smith et al., 2020a). High flow events primarily occur during the winter months due to more frequent low-intensity rainfall, lower ET, and wetter soils. Despite this, low runoff coefficients are common due to sandy soils limiting rapid lateral flow to only the most compacted agricultural areas, saturated wetlands and sealed surfaces (Smith et al., 2020a). The streamflow is very low during dry summer periods, with flow cessation occurring more frequently and for longer durations since 2013 (Kleine et al., 2020).
Dry summer periods are characterised by the relatively high ET which limits annual groundwater recharge to winter months under forests (Smith et al., 2020b).





### 2.4 Isotopic data collection and analysis

Bulk water sample collection of precipitation and streams (Fig. 1a) was used for deuterium ($^2$H) and oxygen-18 ($^{18}$O) analysis. Daily bulk precipitation sampling began in mid-2018 with an autosampler (Fig. 1a). Stream isotope sample

collection began at the beginning of 2018 as grab samples every second week at three locations (Peat North, Peat South, Demnitz Mill; Fig. 1a) during periods of streamflow. Daily stream sampling at Bruchmill (Fig. 1a) began at the end of 2018 using an autosampler during periods of streamflow. Evaporation was prevented by a layer of paraffin in all autosampler bottles. Bulk soil samples were collected monthly and soil water isotope composition analysed with the direct-equilibrium method (Kleine et al., 2020).

Bulk samples of precipitation, stream, and soil water were analysed in the IGB laboratory with a Picarro L-2130i cavity ring down water isotope analyser (Picarro, Inc., Santa Clara, CA, USA). Samples were standardized against Vienna Standard Mean Ocean Water (VSMOW2) and are presented in δ notation.

A synthetic dataset of isotopes in precipitation was created for the period prior to sampling (Table 1). This was based on the nearest local long-term $\delta^2$H monthly precipitation samples from Tempelhof in Berlin (GNIP; IAEA/WMO, 2020). Monthly

data were correlated against temperature and precipitation amounts, with the correlations used to randomly generate daily $\delta^2$H values (cf. Dehaspe et al., 2018). Random generations were repeated to minimize the difference between the synthetic amount-weighted $\delta^2$H values and Tempelhof monthly $\delta^2$H data. The $\delta^{18}$O precipitation synthetic dataset was developed using the predictive bounds of the local meteoric water line of the DMC to correlate $\delta^2$H to $\delta^{18}$O and generate variability.

### 3 EcH$_2$O-iso model set-up

The EcH$_2$O distributed ecohydrological model integrates components to simulate energy and water balance, carbon uptake, and vegetation dynamics. The model is designed to be forced with inputs from regional climate models (Maneta and Silverman, 2013). EcH$_2$O was coupled with an isotope and water age module (EcH$_2$O-iso; Kuppel et al., 2018) to track $\delta^2$H and $\delta^{18}$O and estimate water ages in each model storage and flux. Here, we present an overview of the components of the water-energy-tracer flux- storage interactions that are relevant for the interpretation of results reported in this paper; a

conceptual diagram of the storage and fluxes for energy and water balance is shown in Fig. S1, with complete details of EcH$_2$O and EcH$_2$O-iso provided by Maneta and Silverman (2013) and Kuppel et al. (2018), respectively.

### 3.1 EcH$_2$O-iso energy balance

The energy balance of each model cell is solved for two layers (canopy and surface) and is driven by incoming shortwave and longwave radiation, as well as air temperature, relative humidity, and wind speed. The canopy energy balance resolves

the effective canopy temperature that balances available radiative energy (net radiation), LE of interception and transpiration, and sensible heat exchanges. The model assumes that variations in canopy heat storage are negligible. The canopy energy balance is very sensitive to the availability of intercepted water ($CWS_{max}$, Table 3) for evaporation, the attenuation of





radiation through the canopy ($K_{beer}$, Table 3), and the environmental constraints that limit transpiration as implemented in a Jarvis-type stomatal conductance model (soil moisture, vapour pressure deficit, light, and temperature). The stomatal

conductance model is dependent on the maximum physiological stomatal conductance of leaf water to the atmosphere ($g_{s,max}$, Table 3). Stomatal conductance is limited when vapour pressure deficit is high, with $g_{s,vpd}$ (Table 3) controlling how sensitive the vegetation is to vapour deficit (low value decrease stomatal sensitivity). Similarly, light conditions lower than optimal vegetation light requirements ($g_{s,light}$, Table 3) limit the stomatal conductance of the vegetation.

The surface energy balance resolves the surface temperature that balances surface net radiation with latent heat, sensible

heat, snowpack heat, and ground heat exchanges. Unlike the balance for the canopy, energy storage variations in the snowpack and soil are important to accurately simulate snowmelt and effective soil temperatures and are taken into account in the solution of the surface energy balance. A new channel evaporation component was recently added and is solved using the same approach, estimating channel surface roughness ($R_{chan}$, Table 3), but neglecting heat storage components and ground flux exchanges.

**3.2 EcH$_2$O-iso water balance**

The water balance in EcH$_2$O-iso also uses a multi-layered top-down approach, with canopy, surface, and three sub-surface (layers 1-3) storages (Fig. S1). Incoming precipitation is intercepted by vegetation. Interception amount, limited by leaf area index (LAI) and a specific leaf water storage parameter, controls the canopy water storage and throughfall. Throughfall and direct incident precipitation accumulates on the soil surface and infiltrates into soil layer 1 using the Green-Ampt model and

Brook Corey parameter ($\lambda_{BC}$, Table 3) and air entry pressure ($\Psi_{ae}$, Table 3) (Te Chow, 2010). Infiltration excess is routed laterally as overland flow as described below. Water infiltrated into the soil is vertically redistributed from the topsoil layer to lower layers using a gravitational drainage model. Downward fluxes start when soil moisture exceeds field capacity at a rate driven by the vertical effective hydraulic conductivity ($K_{eff}$ and $K_vK_{eff}$, Table 3), which increases linearly from zero at field capacity to saturated hydraulic conductivity when the layer is at saturation. Upward water redistribution can occur as

storage excess when lower layers are fully saturated. Water can be extracted from the soil from the topsoil layer as evaporation, and as transpiration from any layer as a function of the proportions of roots contained in the layer. Water can also exit the soil profile as leakance to bedrock (or deeper groundwater) in layer 3 ($L$, Table 3). Return flow to the surface occurs when the entire soil profile is saturated and excess storage reaches the surface. Return flow is routed laterally as surface runoff.

Surface runoff, streamflow, and groundwater flow in the bottommost soil layer are the lateral fluxes that are the three main mechanisms of later water redistribution. Water above field capacity in layer 3 of the soil is allowed to move laterally to the downstream cell using a linear kinematic model driven by the cell slope. Surface runoff is generated from infiltration excess and return flows at the end of each time-step. Overland flow is routed following a steepest descent approach until it reaches the channel and allows reinfiltration at every pixel along the flow path. The model assumes that overland flow generated at

the end of the time step at any given pixel reaches the channel if it is not reinfiltrated along the flow path. Once water is in



the channel it is routed toward the outlet using a non-linear kinematic wave model using a scaled Manning's n value (*Mn*, Table 3) to attenuate channel water.

## 3.3 Water ages and isotopic mixing and fractionation

The isotopic composition and water ages in channel storage and each subsurface store are estimated using a complete mixing assumption (Kuppel et al., 2018) by which inflow is completely mixed with storage. Outflow isotopic composition and water age from each storage are equal to the storage. Fractionation of $\delta^2H$ and $\delta^{18}O$ in soils is conducted using the correction of relative humidity(Lee and Pielke, 1992), kinetic fractionation factor (Mathieu and Bariac, 1996; Braud et al., 2005), and the Vogt (1976) kinematic diffusion value. Soil relative humidity is corrected with a sigmoidal function based on the ratio of soil moisture to field capacity. The kinetic fractionation factor is corrected using soil saturation to adjust the n value (liquid-vapour turbulence) between $n = 1$ (dry soil) and $n = 0.5$ (fully saturated soils). Open water (channel) fractionation is conducted with atmospheric relative humidity and open water kinetic fractionation factors. Given the decadal timescales of groundwater flow in the study region (see Massmann et al., 2009) we used mean residence time (MRT) estimations from groundwater volume (V) and flux (O, MRT = V/O) estimates in the model. The MRT formulation assumes continuous and equal mixing of water in storage, similar to the mixing processes invoked in the EcH$_2$O-iso water age module (Kuppel et al., 2018).

## 3.4 EcH$_2$O-iso model set-up and parameterisation

The model was set-up on daily time-steps for four resolutions, with squared cells of 250, 500, 750, and 1000 m of length. The model was run between January 1, 2007, and December 31, 2019, using the first two years as spin-up. To reduce the effect of the spatial resolution of climate model forcing data on model results (e.g. Liang et al., 2004), forcing data were included as five local climate zones (Table 1). The area of each zone was defined using the distance of the climate station and the Thiessen polygon method. Isotopes in precipitation were applied uniformly across the catchment as limited spatial differences in isotopic compositions were observed. Averaged 8-day LAI values were used to improve the estimation of interception capacity through all seasons. Spatial and temporal patterns of LAI were determined using MODIS data (Table 1) with the upper limit of the croplands and pasturelands corrected using ground measurements and other nearby studies (Wegehenkel et al., 2017; Drastig et al., 2019). Soil and vegetation maps were initialized for the highest (250 m) cell resolution, consolidating soil and vegetation percentages with increasing cell size to keep the same proportion of soil to vegetation for each resolution. Soil parameters for each cell were weight-averaged using the proportion of each soil type (brown earth, podzol, peat, gley; Fig. 1b). A proportion-weighted geometric mean was used for soil conductivity and anisotropy (Sanchez-Vila et al., 2006; Bizhanimanzar et al., 2020). The soil water leakance parameter was non-zero to modulate interactions between the deeper regional groundwater system (not modelled) and the shallower groundwater system (modelled). Soil and groundwater isotopic compositions were initialized using soil and groundwater measurements in 2018 and 2019.





### 3.5 Model evaluation, calibration, and validation

#### 3.5.1 Evaluation

The model was evaluated using two efficiency criteria, the Nash-Sutcliffe efficiency (NSE, Nash and Sutcliffe, 1970) and the normalized root mean square error (NRMSE). Discharge and soil moisture in layer 1 were evaluated at two locations (Fig. 1a), and ET and LE at three locations using NSE. The first soil moisture site was evaluated at Forest Site A (herein referred to as Forest A), which is typical of a managed mixed forest over podzolic soils in the DMC (Smith et al., 2020b; Kleine et al., 2020). The second soil moisture site was evaluated at Alt Madlitz (herein referred to as cropland), which has similar soil

(brown earth) and vegetation (croplands) to the northern reaches of the catchment. A fusion of measured soil moisture and estimated soil moisture for the ERA5 reanalysis datasets were used to calibrate soil moisture at each site. All stream isotope (four locations), soil isotope (one location), groundwater isotope (two locations) and transpiration (one location) simulations were evaluated using the NRMSE. NRMSE was used due to inconsistent time-steps of data collection while emphasising dataset variability.

Correlations between fluxes, storages, water ages, and the proportion of vegetation and soils (i.e. spatial proportions in Fig. 1b, c) were assessed using the Spearman's rank correlation (Supplementary Material E). The Spearman's rank correlation was used as it does not assume a normal distribution. Significance of the correlations was assessed to 95% confidence.

#### 3.5.2 Sensitivity analysis

Sensitivity analysis in the DMC was conducted for each model resolution using a modification of the Morris method
(Morris, 1991; Sohier et al., 2014). This is a step-wise sensitivity test, changing model parameters one at a time and quantifying the resulting magnitude of change in model output. Parameter sensitivity was assessed using 75 trajectories with randomized initial parameters (Latin Hypercube Sampling, LHS; McKay et al., 1979), to establish a synthetic baseline. Radial sampling was utilized in a step-wise manner, varying each parameter by 50% of the range. All possible model parameters were included to identify the most sensitive parameters for use in calibration. Output time-series (ET, LE,
discharge, and soil moisture) were evaluated against the synthetic baseline using the root mean square error (RMSE). The RMSE of output for each trajectory was averaged to give an overall parameter sensitivity.

#### 3.5.3 Calibration

Using the most sensitive parameters identified by the analysis, 100,000 parameter sets were generated for Monte Carlo simulations using LHS to optimize sampling space. As the parameter ranges were set the same for all spatial resolutions, the
same parameter sets (100,000) were used. Model testing revealed that two years of spin-up (January 2007 – December 2008) were sufficient to initialize soil moisture storage, groundwater, and discharge. Initial conditions for water ages in storages were determined using previous estimates of shallow soil water (Smith et al., 2020b) and nearby tritium groundwater age estimates (Massmann et al., 2009). Regression of water ages time-series was conducted (p-value < 0.05) to ensure that no





significant long-term change in water ages was present. Model calibration was conducted with a discontinuous period, 2009
– 2014 and 2018 – 2019, with the 2015 – 2017 years used for validation. The calibration period was selected due to a
combination of high and low flow extreme events and data availability. The calibration extent was limited to the Demnitz
Mill sub-catchment (Fig. 1b and c) due to the strong regional groundwater interaction with surface water at the outlet of the
DMC (Smith et al., 2020a). Multi-criteria calibration (Section 3.5.1) was conducted using normalized efficiency criteria in
empirical cumulative distribution functions to rank the best overall efficiency (Supplementary Material B; Ala-aho et al.,
2017; Smith et al., 2020c). Posterior parameter ranges of calibrated parameters are provided in Table S2 and Fig. S3. Single
calibration was conducted for each model output to directly compare the effect of multi-criteria calibration on model output
trade-off. The significance of the difference of efficiency criteria at each resolution was assessed using the Wilcoxon rank-
sum test (Mann and Whitney, 1947). The Wilcoxon rank-sum test does not assume a distribution and is, therefore, more
robust in the comparison of efficiency criteria.

### 3.5.4 Validation

Model validation was conducted for years not used for calibration (2015 – 2017). The validation years had average flow
conditions relative to the long-term measurement and were, therefore, representative of the average conditions of the
catchment. The model was validated against measured discharge at Demnitz Mill, remotely sensed ET and LE, and soil
moisture estimated from ERA5 reanalysis products at the same sites as calibration (Table S3). Since isotopic measurements
(stream, soil, and groundwater) began in 2018, isotopic data were not available for validation. "Soft" validation was assessed
using soil isotopes not used in calibration (soil layer 2 and cropland isotopes layer 1) in 2018-2019.

### 4 Results

### 4.1 Sensitivity to spatial model resolution

The ranked sensitivity of model output (standardized RMSE between 0 and 1 for maximum and minimum of all resolutions)
against all model parameters (18, 30, 6, parameters for each soil, vegetation, channel, respectively) showed that the RMSE of
model output is sensitive to very few parameters which control the dominant fluxes (Fig. 2a). The few sensitive parameters
resulted in the selection of a much smaller number of calibrated parameters (Supplementary Material B; 10, 6, 4, parameters
for each soil, vegetation, channel, respectively). Regardless of the calibration against remote-sensed products (ET, LE), field
data (discharge), or fusion of data sources (soil moisture), results showed high non-linearity against the ranked parameters
(low average sensitivity to high average sensitivity). Each grid resolution showed similar non-linearity of RMSE to
parameters. Splitting the ranked parameters into the vegetation and soil parameters isolated their contribution to the
sensitivity of each output (Fig. 2b and c). The standardized RMSE showed higher sensitivity of parameters for all outputs
when the resolution was finer (Fig. 2b and c). Specifically, greater separation of sensitivity was present in the vegetation
parameters mainly influencing ET (Fig. 2b) and soil parameters regulating soil moisture and runoff generation (Fig. 2c), with





the largest change with resolution occurring between 500 to 750 m. Latent heat and soil moisture in layer 2 did not show differences in parameter sensitivity between resolutions for either vegetation or soil parameters, underlining the importance of layer 1 in water partitioning.

The output for calibration to remote-sensed products (ET and LE) was most sensitive to vegetation parameters (Fig. 2b), particularly canopy water storage ($CWS_{max}$) and maximum stomatal conductance ($g_{s,max}$). Large parameter ranges in $CWS_{max}$
resulted in high variation in LE, and thereby ET and interception evaporation (not shown). At all resolutions, stream discharge was sensitive to three parameters: Manning's n ($Mn$), horizontal saturated hydraulic conductivity ($K_{eff}$), and vertical to horizontal hydraulic conductivity anisotropy ($K_vK_{eff}$).

## 4.2 Effects of resolution on calibration

The values of the median calibration efficiency criteria of the 100 "best" parameter sets for each model output and resolution
suggest dominant catchment processes were reasonably captured (Table 4). Median validation efficiencies generally showed small decreases compared to the calibration period (Table S3). Multi-criteria calibration showed different trade-offs in efficiency between resolutions (Table 4) with the maximum model efficiency (i.e. single model calibration; Table S1) not simultaneously met. Except for ET and LE (calibration to remote-sensed data), the model performance was substantially better at finer resolutions (Table 4, Table S1). While simulations of soil moisture displayed relatively high single calibration
efficiency (Table S1), multi-criteria calibration resulted in lower model performance (Table 4).

Field data had the greatest benefit for constraining results at finer resolutions, most notably with significant improvements in discharge and stream isotopes (Table 4). Additionally, transpiration dynamics (in the mixed forest) were greatly improved at 250 m relative to the other resolutions, despite similar vegetation percentages at the location for all resolutions (Fig. 1c). Similarly, a greater capability to simulate soil moisture was apparent at finer resolutions. However, significant improvements
in soil moisture with decreasing resolution were not consistent.

Model output calibrated against remote-sensed data also showed mixed patterns of model performance between resolutions (Table 4). In general, coarser resolutions performed better against remote sensed data than finer resolutions through both single calibration (Table S1) and multi-criteria calibration (Table 4).

The modelled water balance fluxes were quite similar across scales (Table 5). The largest contribution of precipitation loss
within the catchment was ET, accounting for >80 % of total precipitation. Tr was the dominant component of ET, accounting for ~50% of losses, with interception evaporation (Ei: 21-25%) and soil evaporation (Es: 9-12%) much smaller. Secondary outflows of the catchment were stream discharge (11-14%) and vertical groundwater leakance (2-4%) to the deeper regional aquifer (Table 5).

## 4.3 Resolution effects on estimations of discharge and stream isotopes

All model resolutions were able to adequately simulate discharge at both Demnitz and Demnitz Mill, with minor improvements in low flows at coarser resolutions, and improvements in high flows at finer resolutions (Fig. 3e and f).





Uncertainty was also lower with finer grids. Isotopic simulations in the northern reaches of the catchment were constant and relatively similar between resolutions (Fig. 3a). More notable deviations of median simulations between resolutions were evident at stream sites downstream of the wetland between Peat North and Peat South (Fig. 1a, Fig 3b-d); with failure to

reproduce winter depletion and summer enriched isotopes at resolutions >500m. While enrichment of in-stream isotopes could be reproduced for single calibration at coarser resolutions (Table S1), multi-criteria calibration was unable to capture isotopic enrichment simultaneously at all downstream sites. Multi-criteria calibration resulted in a wide range of simulated in-stream isotopic compositions at coarser resolutions (750 and 1000 m, Fig. 3a-d), consistent with simulations of spatially extensive overland flow events not present at finer resolutions. The range of upper and lower bounds of simulated stream

isotopes increased notably between Peat North and Peat South, as a result of both the uncertainty of process representation and wetland open water fractionation within the wetlands (Fig. 3a-b). These uncertainties were primarily within the wetlands, with the range of upper and lower bounds decreasing with distance from Peat South (Fig. 3c-d). For all stream isotope locations, the range of upper and lower bounds decreased with coarser model resolution.

**4.4 Resolution effects on estimations of discharge and stream isotopes**

Calibration of ET and its partitioning (i.e. transpiration, Tr) used data fusion of remote-sensed ET (MODIS), and field measurements of sapflow at Forest A. Calibration of ET to the 8-day MODIS ET showed a small (19 mm) increase in the median annual catchment ET estimated by coarser resolutions (Fig. 4a-d) but with increased uncertainty. This difference between resolutions was small in comparison to the uncertainty of annual ET. Spatially, ET had a positive correlation with coniferous forest cover and peaty and podzolic soil cover at most resolutions, and a consistent negative relationship to the

proportion of pastureland and brown earth soils (spatial comparison of Fig. 1 and Fig. 4, statistical correlations shown on Fig. S5). The fraction of Tr to ET (Fig. 4e-h) was also relatively consistent between resolutions, with only a slight decrease at coarser scales. The uncertainty of the ratio of Tr to ET did not notably change between resolutions. Unlike ET, Tr had strong dependencies to both vegetation and soil proportions at all resolutions (Fig. 1; Fig. 4; Fig. S5). Specifically, Tr strongly increased with higher proportions of croplands and brown earth, and strongly decreased with higher proportions of

conifer and pasturelands and peaty and podzolic soils. Like ET, the ratio of Es to ET increased moderately with the model resolution, but the increase was still within the model uncertainty. Es showed a much weaker dependency to soil or vegetation proportion with only higher proportions of pastureland and peaty soil significantly increasing Es. Median annual channel evaporation was relatively constant from 250 to 750 m resolutions, with a slight decrease at the 1000 m resolution (Fig. 4m-p). Channel evaporation periodically resulted in dry channels and a discontinuous channel network during dry

periods (not shown), which was consistent with stream connectivity observations in the field. The annual ratio of recharge (Re) to ET, and the ratio uncertainty, was consistent across all resolutions (Fig. 4q-t). Spatially, recharge was closely linked to soil cover, with a moderately positive correlation with brown earth and moderately negative correlations with peaty and podzolic soils. However, the proportions of conifers showed the strongest linkage with annual recharge, which greatly





decreased with higher proportions of conifers. The decrease in annual recharge is largely linked to the higher ET, mainly
from high interception losses (Fig. 1; Fig. 4; Fig. S5).

Calibration of soil moisture in layer 1 (against measured data and ERA5-reanalysis) in the cropland and forest sites provided
adequate representation of the measured dynamics (Fig. 5a&b, respectively). All model resolutions showed slight over-
wetting of the soils in both layers 1 and 2 during the summer months (June – August, Fig. 5a-d), with simulations closer to
ERA5-reanalysis than measured soil moisture. Except for the 500 m grid, the range of the upper and lower bound in soil
moisture decreased with coarser model resolution. The dynamics of simulated soil isotopes in the forest (calibrated) and
grassland (validated) captured measurements at all resolutions and depths, with a slight increase in enrichment with
increasing resolution (Fig. 5e-h). The ranges in upper and lower simulation bounds were notably smaller in the 250 m
resolution during periods of measurement; however, the ranges in simulation bounds were more similar when calibration
data were not available. The model adequately reproduced the relatively stable field measurements of groundwater isotopes
in wells 4 and 8 with limited differences in either the median or simulation bounds between resolutions (Fig. 5 i&j).

Annual median equivalent soil water depths in layers 1 - 3 were relatively consistent for each model resolution (Fig. 6). The
uncertainty of equivalent water depth in layers 1 and 2 was small relative to layer 3 due to non-calibrated soil depths in the
first two soil layers (Supplementary Material B). Higher proportions of pastureland and peaty soil strongly increased soil
storage in layers 1 and 2, and consequently greatly increased soil evaporation (Fig. 1; Fig. 6; Fig. S5). There was a weak
negative relationship of soil storage with croplands, with a strong negative relationship with conifers; however, the
dependency of soil storage with conifers was not consistent for all resolutions. Layer 3 storage had much stronger (negative)
correlations with both vegetation (conifers) and soils (brown earth and podzolic soils) for all resolutions than the upper soil
layers. Notably, all storages decreased over the time of the simulations (2009-2019) mainly as a result of the 2018-19
drought (Table 5).

**4.5 Ages of ecohydrological fluxes**

There was a notably lower estimated shallow soil water age (layers 1 and 2) at the 250 m model resolution relative to all
coarser model resolutions (shown for summer months, Fig. 7a-h, seasonal ages in Table S4). Uncertainty of water age
estimates in each layer also generally decreased with coarser resolutions; trivial in layer 1 but more notable in layer 2. Water
ages in layers 1 and 2 strongly decreased with increased Tr, Re, and higher proportions of croplands, and brown earth.
Conversely, water ages (layers 1 and 2) strongly increased with higher proportions of conifers, peaty and podzolic soils for
all model resolutions (Fig. 1; Fig. 7; Fig. S5).

Unlike layers 1 and 2, the median and uncertainty of modelled Tr and groundwater (GW) ages increased with model
resolution (Fig. 7i-p). On average, Tr water ages were slightly older than one year. GW ages were decadal at all resolutions,
with an increase in GW age in the mid-reaches of the catchment. Spatial proximity of groundwater storages to streams was
apparent at finer resolutions (Fig. 7m&n), with decreasing riparian GW ages in cells with streams. Tr water ages were much
older with higher ET and much younger with higher Re (Fig. 4; Fig. 7; Fig. S5). A similar strong increase of Tr age was





observed with proportions of conifer, peaty and podzolic soils, and a strong decrease with croplands and brown earth proportions. Dependencies of GW ages to fluxes, storage, vegetation and soils at all model resolutions were relatively limited. A consistent correlation of GW age was observed with Re (strongly negative), the proportion of conifer forest
(strongly positive), and gley soil (weakly negative).

To characterise stream water ages during low, medium, and high flow conditions, water ages were averaged for different flow conditions ($Q_a = (Q - \bar{Q})/\sigma$, $\bar{Q}$ is mean discharge, σ is the standard deviation). High flow was defined as $Q_a > 1.0$, and low flow $Q_a < -0.5$. Similar to GW, modelled stream water ages and uncertainty increased with finer resolution (Table 6); however, stream water ages were notably younger than GW. At resolutions >250 m, stream water ages under all flow
conditions moderately decreased from the headwaters (Peat North) to Demnitz Mill, whereas stream water ages increased downstream at the 250 m resolution. For all resolutions, high flow conditions showed a notable decrease in stream water ages compared to the medium and low flow conditions (Table 6). During the largest events, stream water ages dropped most notably in the 750 and 1000 m resolutions (average stream water age of 0.5 years during large events) reflecting extensive overland flow simulations (Table 6). Stream water ages for the finer grids also decreased during large events  (average
stream water age of 1.8 years during large events); however, the change was not quite as large relative to the long-term average stream water age compared to coarse grids.

## 5 Discussion

### 5.1 Utility of tracer-aided ecohydrological models in constraining water storage-flux-age interactions at different spatial resolutions

This study demonstrated the effectiveness of a physically-based, tracer-aided ecohydrological model in consistently simulating blue and green water fluxes, and their relationships to vegetation, soil cover and water storage in a 66 km$^2$ catchment across different spatial resolutions. The relatively minor variability of catchment scale fluxes between model resolutions is promising for the continued development and use of tracer-aided ecohydrological models at larger scales and spatial resolutions under a wide range of hydroclimatic conditions, including extreme droughts, where changes in blue and
green flux partitioning can be marked (Prudhomme et al., 2014). Furthermore, the catchment-wide water balance and water loss, including leakance to the regional groundwater system, was consistent with simpler monthly water-balance estimates for the catchment (Smith et al., 2020a). The larger areas of more homogenous landcover (e.g. croplands and conifer forests) consistently showed spatial patterns (Figs. 1 and 4) between model resolutions of lower ET and higher recharge in the croplands, and a higher ET, a lower ratio of Tr to ET (due to interception), and a lower recharge in the conifer forests. These
catchment scale results are consistent with findings of previous plot-scale studies in the region (Douinot et al., 2019; Smith et al., 2020b; Kleine et al., 2020). Similarly, parameterizations of the rooting zone were consistent across all resolutions for each vegetation type (Fig. S4), suggesting the calibration achieved an effective constraint for more consistent, larger-scale modelling of ecohydrological interactions. Model results for the wetlands in the mid-reaches of the catchment captured





wet/saturated conditions across all resolutions, despite no direct soil moisture calibration in this area (Fig. 6). This suggests
sufficient model skill to reproduce differences in moisture conditions where slight variations in topography cause marked
ecohydrological contrasts.

One of the main advantages of tracer-aided models is capturing the distributions and dynamics of water storage involved in
mixing processes that explain the damping and lagging of precipitation signals being transmitted through the system. At the
catchment-scale, all model resolutions reproduced a groundwater-dominated stream flow system that is driven by recharge in
the headwaters. That recharge is very sensitive to land cover and ecohydrological partitioning. As a result, the modelled
catchment flow domain links relatively large, rapidly circulating/recharged sources of groundwater in the slightly steeper
headwaters under crops, to shallower stores of older groundwater under forests that receive much lower recharge (Figs. 6 and
7). Near-surface water storage is greatest in the wetland areas where younger water can contribute to streamflow as localised
overland flow in wet periods; however, this was only reproduced at the two finer model resolutions. Thus, while the
dynamics and totals of blue and green fluxes and water storage-flux-age interactions of dominant vegetation and soils (e.g.
croplands/brown earth, conifers/podzols) were relatively consistent between model resolutions (Fig. 4), there were
differences for localised hydrologically-important vegetation-soils grids at different grid scales. This probably reflects
calibration not capturing subtle, but important differences in the representations of modelled flow paths at coarser resolutions
(Blöschl and Sivapalan, 1995; Harvey, 2000; Vereecken et al., 2007). For example, limited variability of stream isotopes,
older groundwater and stream water ages observed at model resolutions greater than 500 m are consistent with a loss of
smaller-scale process representation (Fig. 3 and 7).

These latter differences in process representation likely reflect interactions in the deeper soils and between groundwater and
the stream network, as water storage and ages at all resolutions in layers 1 and 2, as well as soil evaporation ages, were
consistent with those estimated in other similar catchments (e.g. Douinot et al., 2019) and plot scale modelling in the DMC
(Smith et al., 2020b).

Modelled groundwater ages at finer resolutions (250 and 500 m) were more consistent with local groundwater tritium age
dating, and similarly showed a decrease in groundwater age in closer proximity to stream channels (Massmann et al., 2009).
In contrast, coarser resolutions produced older, more spatially uniform water ages. Finer model resolutions additionally
showed spatial patterns of groundwater inflows to streams (not shown) in locations that were consistent with the historical
distribution of wetlands and ponded areas prior to drainage (Gelbrecht et al., 2005). This suggests a dissociation of some
important hydrological processes in riparian areas at coarser scales, including localised overland flow in wetlands which
contributes to runoff peaks in winter and associated isotope variations (e.g. Grabs et al., 2012). This probably contributed to
the loss of tracer dynamics as aggregation more coarsely represents storage-flux-age interaction due to averaging of spatial
heterogeneity of vegetation and soil/subsurface properties (Ershadi et al., 2013; Yang et al., 2001).





### 5.2 How do coarser resolutions affect robustness and uncertainty of model parameterisation, sensitivity, and calibration?

Changes in parameter sensitivity and posterior parameter ranges with model resolution provides key information on internal process representation that can help reduce the degrees of freedom and increase model robustness (Blöschl and Sivapalan, 1995). The broadly consistent ranking of sensitive parameters for each resolution (Fig. 2) suggests that the model parameters performed similarly across different grid scales, albeit with slightly less sensitivity for individual parameters at coarser resolutions. Discharge showed the most notable decrease in sensitivity across resolutions, driven by the scaling of Manning's n (Fig. S3). The sensitivity of Manning's n across resolutions is non-linear, accounting for changes in numerical wave-routing (Courant criteria), channel length, roughness, and shape in larger resolutions (similar to Bhaskar et al., 2015). In soils (soil moisture), deviations in parameter sensitivity and uncertainty across resolutions are likely due to the aggregation of topographic features (e.g. slope) which can change the details of groundwater movement and distribution of saturation areas surrounding the stream (e.g. Yang et al., 2001). Vegetation properties generally control the sensitivity of the energy balance (particularly LE) which is unsurprising given the dependence of remote-sensed LE estimations on vegetation coverage (e.g. Mu et al., 2011). The low sensitivity of energy balance components to soils is likely due to low soil evaporation (soil latent heat) and outgoing shortwave (low albedo) contributions.

Evaluation of catchment flux-storage-age representations with finer resolutions resulted in significantly better model efficiencies (average 20% improvement with the 250 m resolution compared to all others), with an apparent threshold between 250 m and 500 m (Fig. S2). The better model efficiency at finer resolutions may be due to either a spatial threshold of calibration data (e.g. measured soil moisture) for coarse resolutions (limiting efficiency) or a loss of dynamics in larger grids due to aggregation of fine-scale landscape characteristics (Samaniego et al., 2017). Additionally, the lower efficiency at coarser scales (e.g. stream isotopes) may be due to trade-offs between optimizing poor-fit outputs (especially stream isotopes) and well-fit outputs in multi-criteria calibration (Efstratiadis and Koutsoyiannis, 2010). However, it should be noted, that even simulations of general isotope values in streamflow, despite the loss of short-term dynamics, is still likely much more indicative of reasonable catchment scale storage-flux estimates than a model not using tracers (Birkel et al., 2011; Holmes et al., 2020).

As with efficiency, uncertainty, in terms of the range of upper and lower simulation bounds, improved (decreased) with finer resolutions. In addition to wider simulation bounds at calibrated sites, coarser scales revealed larger variability, particularly in stream isotope and water age simulations. The large variability suggests, consistent with comments in 5.1 above, that key processes (i.e. local overland flow and groundwater fluxes) and isotopic mixing (wetland isotopic mixing) are not well constrained (Tetzlaff et al., 2017; Kuppel et al., 2018). The increased uncertainty and variability is particularly notable in the groundwater age for coarse resolutions, which is more than four times larger than finer resolutions (Fig. 7). The high uncertainty at coarser scales complicates evaluation of the storage to buffer climatic change (Kløve et al., 2014) or water quality impacts (Hill, 2019). However, the primary differences in uncertainty between model resolutions appeared to be





mainly restricted to isotope and water age estimations, as catchment average variability were similar across resolutions for ecohydrological fluxes (e.g. Figs. 4, and 6).

## 5.3 Implications for large scale modelling vis-à-vis the use of field data, remote sensing data (e.g. MODIS or reanalysis data), or data fusion


Evaluation of how ecohydrological fluxes, storages, and water ages and their uncertainties upscale from smaller nested field sites to larger scales is useful for coupling is with regional climate models and is essential for science to inform management, as policy focuses on larger scales with broader implications on societal water demands and usage (Asbjornsen et al., 2011).

To better understand how well feedback mechanisms between soil, vegetation, and the atmosphere are captured in modelling at various scales, long-term, multi-scale data collection is required particularly for soil moisture due to the strong influence of vegetation on soil moisture dynamics (Asbjornsen et al., 2011; Vereecken et al., 2019). Here, multiple soil moisture measurements in various soil-vegetation systems were useful in constraining model calibration, with a stronger influence at finer scales (Table 4). In addition, other high-resolution data (e.g. especially spatial resolution for isotopes, but also temporal

resolution for sapflow) proved beneficial for calibration at the catchment scale, highlighting the value of long-term data collection in experimental catchments (Tetzlaff et al., 2017). Stream isotopes were of primary importance in revealing differences in model process fidelity as well as reducing the uncertainty of flow paths representation. In particular, processes in key hydrologic "hot spots" (e.g. 2.8 km$^2$ wetland between Peat North and Peat South) were constrained solely by stream and groundwater isotope field data. Isotopic data likely helped to constrain the calibration especially regarding the

evaporative enrichment and mixing in the wetland (Sprenger et al., 2017). However, there was an apparent scale limit to the value of isotope field data in capturing more localised processes with coarser scales too large to adequately capture important wetland processes. This example suggests an "upscaling-process limit" for isotopic impacts and a Representative Elementary Area between 500 and 750 m for the DMC (Wood et al., 1988; Blöschl and Sivapalan, 1995). For all resolutions, the shorter time-series for water stable isotopes resulted in gaps in key calibration periods and larger uncertainty when data were not

available (Fig. 6e-h). Additionally, the model was able to adequately reproduce stream isotopes at a single location (Table S1); however, processes were only fully constrained with the multi-criteria calibration of all stream isotope locations. These data limitations highlight the value of spatially distributed long-term data to drive model improvements (Soulsby et al., 2015).

These inevitable limitations in the availability of long-term data emphasise the inherent advantages of remotely sensed data.

While the remote-sensing data was essential for constraining the catchment-wide ET and LE (8-day and annual amounts), the limited heterogeneity in the coarser scale remote-sensed products (Table 1) resulted in a poorer fit at finer resolutions, which had more spatial heterogeneity at key field-scale measurement sites. The lower calibration efficiency criteria of finer resolutions to remotely sensed ET and LE estimations is likely due to scaling in the latter. The relatively coarse grid resolution of the MODIS ET and LE data aggregates multiple land cover types into each grid which is known to result in

large uncertainties in daily and monthly ET estimates (Gowda et al., 2007; Velpuri et al., 2013). Additionally, at finer





resolutions, the lower modelled soil wetness relative to coarser resolutions restricted ET and LE relative to MODIS estimations (i.e. lower annual ET and model ET efficiency), which estimates ET and LE from vegetation coverage (Mu et al., 2011). Lower ET and LE have also been seen at the plot-scale during dry years (Smith et al., 2020b). Of the calibrated vegetation ET, the conifer forests were most different from MODIS data (Table 4), which could be due to a number of

reasons. The dependence of MODIS ET on vegetation coverage has been shown to be influenced by underlying errors in overestimating LAI variability in conifer and broadleaf forest have been identified due to uncertainties in atmospheric correction (Heiskanen et al., 2012), leaf optical characteristics during leaf out and senescence (Wang et al., 2005), and over-compensation of understory canopy development (Jensen et al., 2011). However, these underlying errors and limited dependencies on soil wetness are likely minimized on an annual basis and under average precipitation (e.g. similar

catchment-wide ET between model resolutions, Fig. 4). The fusion of additional products (e.g. Landsat, Sentinel-2) may further alleviate some inter-annual uncertainties of remote sense datasets; however, extreme conditions still require ground-truthing, underlining the need for data-rich experimental catchments.

## 6 Conclusion

Long-term water security is dependent on quantitative knowledge of regional water storages and fluxes and how these are

anticipated to change under the anticipated increased frequency of extreme events, such as droughts (Falkenmark and Rockström, 2006). However, how the vegetation-soil-atmosphere interactions regulating these ecohydrologic fluxes are additionally expected to change is not well known. This is particularly the case at larger scales, resulting in an uncertain evidence base for land-use decision making in regions where water resources are under stress (e.g. agricultural areas) (Falkenmark and Rockström, 2010). The significant challenges at larger scales are tied to the limitations of appropriately

upscaling controlling vegetation-soil interactions in models between spatial scales.

The physically-based tracer-aided ecohydrologic model $EcH_2O$-iso allowed us to assess the effects of spatial interactions across model scales using four resolutions of the same 66 km2 catchment in NE Germany. This used multi-criteria calibration of field data (discharge, soil moisture, stream, and soil isotopes) and remotely sensed data in a data fusion approach. Fluxes and water storages were reproduced similarly across all model resolutions, with the dominant soil and

vegetation covers largely explaining the spatial distributions. Isotopic and water age simulations revealed limitations at larger spatial resolutions for internal mixing mechanisms, most notably surface runoff, wetland evaporation and deeper groundwater mixing. Despite this, for all model scales, spatially distributed datasets of both remote-sensed products as well as more local field data (particularly isotopes) were useful calibration constraints in modelling ecohydrological fluxes, whilst also giving a plausible representation of water storage and age interactions at the catchment scale. The effectiveness of the

model for simultaneously capturing ecohydrologic fluxes, storages, and age interactions for each resolution provides a promising basis for further testing of upscaling spatio-temporal influences of soil-vegetation-atmospheric interactions in larger catchments.





**Acknowledgements**

The authors acknowledge funding from the European Research Council (project GA 335910 VeWa). Contributions from CS
were supported by the Leverhulme Trust through the ISO-LAND project (RPG 2018 375). Isotopic analysis was conducted
by David Dubbert at the Leibniz-Institute of Freshwater Ecology and Inland Fisheries. The authors acknowledge the
University of Aberdeen IT services for the use of the High-Performance Cluster (HPC), which was used for all model runs.

**Code/Data availability**

The model code of EcH$_2$O-iso is publically available at https://bitbucket.org/sylka/ech2o_iso/src/master_2.0/. The data used
are available from the corresponding author upon request.

**Author contribution**

AS conducted model set-up, calibration, and validation with the EcH2O-iso model from earlier work led by DT, CS, and
MM. Data used for the model calibration and validation were collected by LK. All the authors contributed to model
interpretation. AS prepared the manuscript with contributions from all the co-authors. All authors contributed to manuscript
editing.

**Competing interests**

The authors declare that they have no conflict of interest.

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





Figure 1: (a) The location of the Demnitzer Millcreek Catchment (DMC) in Germany and field measurement locations within the DMC for soil moisture (squares), stream (circles), groundwater (stars), and precipitation isotopes (diamonds), (b) Soil coverage of





brown earth, gley, peat, and podzol for spatial resolutions 250, 500, 750, and 1000m (c) Vegetation coverage of broadleaf forests, conifer forests, croplands, and pasture lands for each spatial resolution. Black boundaries show the calibration extent of the
Demnitz Mill sub-catchment.

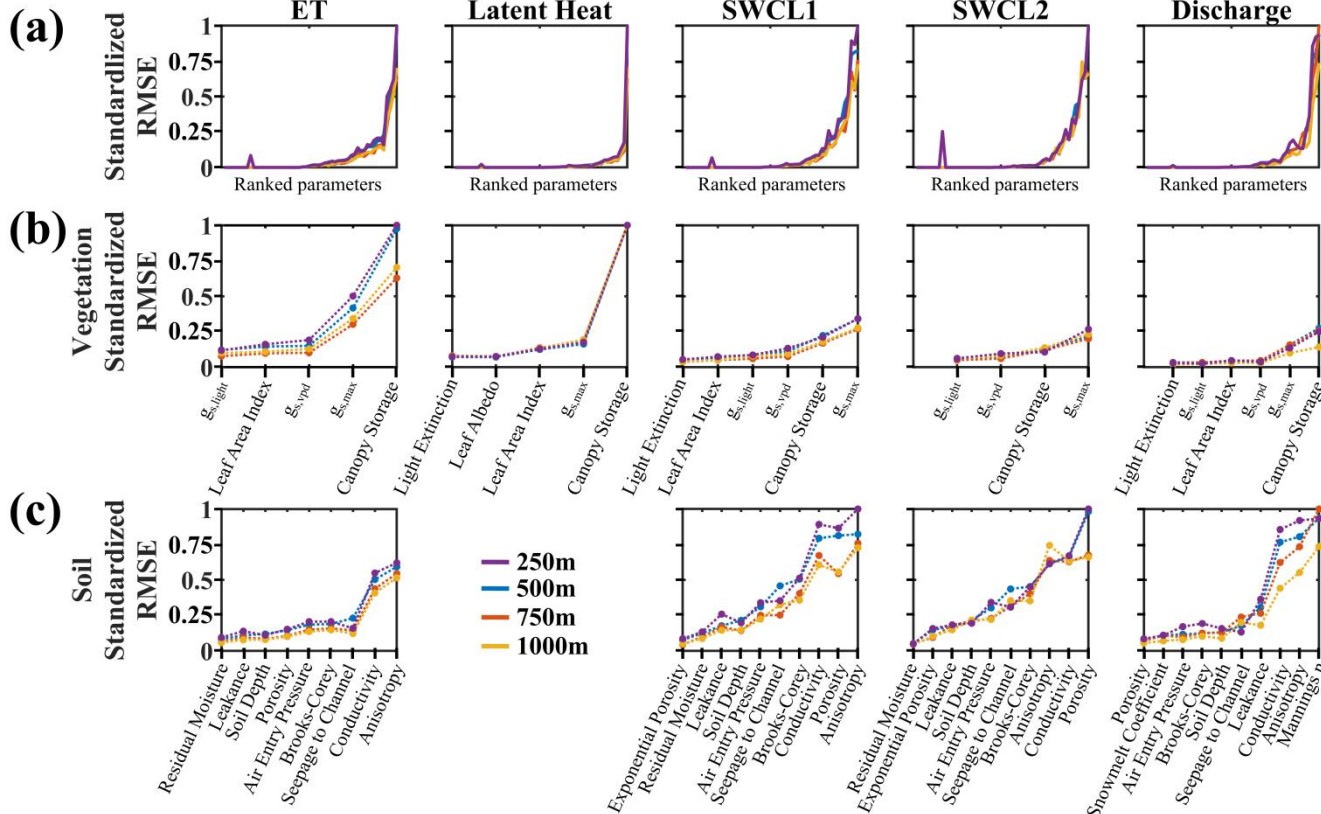

Figure 2: (a) Standardised root mean square error, (b) standardized root mean square error for the most sensitive vegetation parameters, and (c) standardized root mean square error for the most sensitive soil parameters for each output and spatial resolution. Vegetation parameters, gs, are the control of vegetation stomatal conductance, with maximum potential conductance
($gs,max$), light controlling conductance ($gs,light$), and vapour pressure deficit controlling conductance ($gs,vpd$).



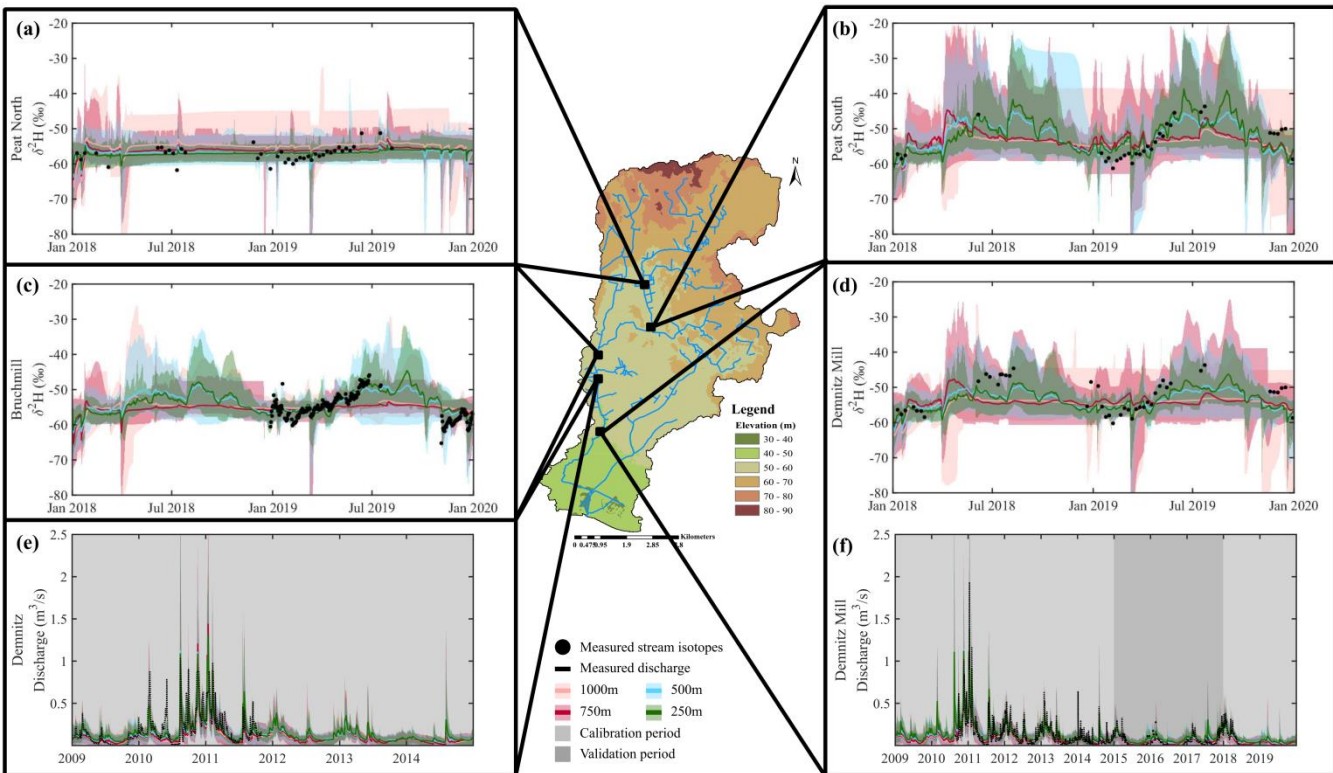

**Figure 3: Simulated stream isotopes at (a) Peat North, (b) Peat South, (c) Bruchmill and (d) Demnitz Mill, and discharge at (e) Demnitz and (f) Demnitz Mill. Each site is shown with reference to its location in the catchment. Colours indicate the model resolution, with solid lines showing the median simulation and shaded regions showing the upper and lower simulation bounds.**





**Figure 4: The (a-d) average annual evapotranspiration (ET), (e-h) ratio of transpiration (Tr) to ET, (i-l) ratio of soil evaporation (Es) to ET, (m-p) channel evaporation (channel E), and (q-t) the recharge proportion of vertical flux (recharge + ET) for the 250, 500, 750 and 1000m resolutions. Black boundaries show the calibration extent of the Demnitz Mill sub-catchment. Values shown are the catchment-wide long-term (2009 – 2019) average values and average standard deviation (average of each pixel) within the Demnitz Mill sub-catchment.**



**Figure 5: Simulations and measured soil moisture in layers 1 (a & b) and layer 2 (c & d) at in the croplands (a & c) and forest (b & d). Also shown are adjusted ERA5 soil moisture estimates at the same locations. Measured and simulated soil isotopes in layer 1 in (d) the forest (e) the cropland, and layer 2 in (g) the forest and (h) the cropland. Simulated and measured groundwater isotopes at (i) groundwater well 4 (GW4), and (j) groundwater well 8 (GW8).**




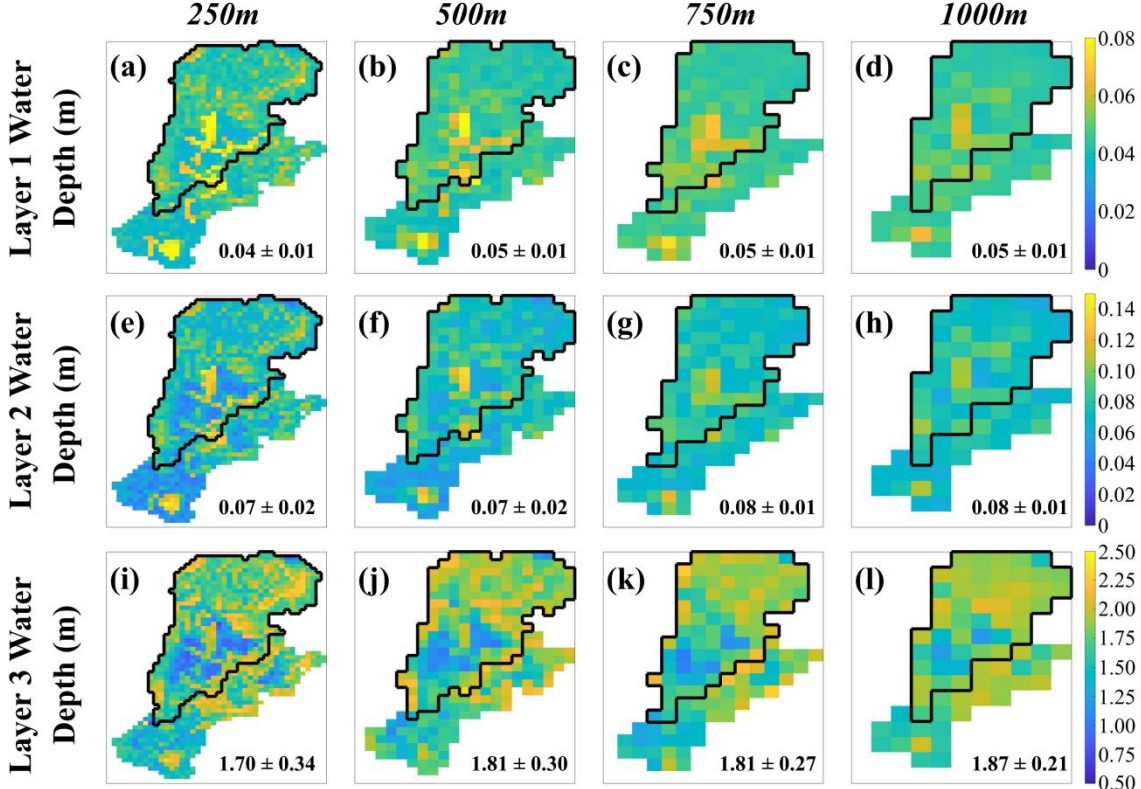

**Figure 6: The (a-d) median annual equivalent water depth in layer 1, (e-h) median annual equivalent water depth in layer 2, (i-l) median annual equivalent water depth in layer 3 for the 250, 500, 750 and 1000m resolutions. Black boundaries show the calibration extent of the Demnitz Mill sub-catchment. Values shown are the catchment-wide long-term (2009 – 2019) median values and average standard deviation (average of pixel standard deviation) within the Demnitz Mill sub-catchment.**



Figure 7: Estimated average water ages during the summer (Jun – Aug) in (a-d) soil layer 1, (e-h) layer 2, and (i-l) transpiration, and (m-p) long-term groundwater mean residence time (GW MRT) for model resolutions 250, 500, 750, and 1000m, respectively. Black boundaries show the calibration extent of the Demnitz Mill sub-catchment. Values shown are the catchment-wide long-term (2009 – 2019) average values and average standard deviation (average of each pixel) within the Demnitz Mill sub-catchment.






**Table 1: Demnitzer Millcreek catchment climate data spatial and temporal resolutions from nearby weather stations, site collection, and remote sensed datasets (ERA5 and MODIS) used for daily modelling.**

| | Forcing Datasets | | | | | | |
|---|---|---|---|---|---|---|---|
| | **Spatial Resolution** | **Temporal Resolution** | **Locations (Latitude and Longitude)** | | | | |
| **Precipitation (m/s)** **Temperature ($^o$C)** **Wind speed (m/s)** **Relative humidity (%)** | N/A | Daily | Lindenberg (52.21$^o$N, 14.12$^o$E) | Manschnow (52.55$^o$N, 14.55$^o$E) | Muncheberg (52.52$^o$N, 14.12$^o$E) | New Madlitz (52.36$^o$N, 14.25$^o$E) | Furstenwalde (52.4$^o$N, 14.1$^o$E) |
| **Short wave radiation (W/m$^2$)** **Longwave radiation (W/m$^2$)** | 500 m | Daily | N/A | | | | |
| **δ$^2$H [‰]** **δ$^{18}$O [‰]** | N/A | Daily (June 2018 - Dec. 2019) | Hasenfelde (52.41$^o$N, 14.18$^o$E) | | | | |
| **Leaf Area Index (m$^2$/m$^2$)** | 500 m | 8 – day | N/A | | | | |
| | Calibration and Validation Datasets | | | | | | |
| | **Spatial Resolution** | **Temporal Resolution** | **Locations (Latitude and Longitude)** | | | | |
| **Discharge** | N/A | Daily (2007 - 2019) | Demnitz Mill (52.37$^o$N, 14.19$^o$E) | | | | |
| | N/A | Daily (2007 - 2011) | Demnitz (53.39$^o$N, 14.20$^o$E) | | | | |
| **Stream Isotopes** | N/A | Bi-weekly (2018 – 2019) | Peat North (52.41$^o$N, 14.22$^o$E) | | Peat South (52.40$^o$N, 14.23$^o$E) | | Demnitz Mill (52.2$^o$N, 14.1$^o$E) |
| | | Daily (2018 – 2019) | Bruchmill (52.39$^o$N, 14.20$^o$E) | | | | |
| **Soil Moisture** | 20, 60, 100 cm | 15 -min (2018 - 2019) | Forest Site A (52.39$^o$N, 14.20$^o$E) | | Cropland (52.37$^o$N, 14.23$^o$E) | | |
| **Soil Isotopes (0 – 20 cm)** | N/A | Monthly (2018 – 2019) | Forest Site A (52.39$^o$N, 14.20$^o$E) | | Cropland and Grassland various sites (52.45$^o$N, 14.23$^o$E; 52.43$^o$N, 14.23$^o$E; 52.43$^o$N, 14.22$^o$E; 52.39$^o$N, 14.27$^o$E) | | |
| **Soil Isotopes (20 – 60 cm)** | N/A | Monthly (2018 – 2019) | Forest Site A (52.39$^o$N, 14.20$^o$E) | | | | |
| **GW Isotopes** | N/A | | Well 4 (52.41 $^o$N, 14.22$^o$E) | | Well 8 (52.40 $^o$N, 14.21$^o$E) | | |
| **Transpiration** | Tree Stand | Hourly (2018) | Forest Site A (52.39$^o$N, 14.20$^o$E) | | | | |
| **Evapotranspiration** | 500 m | 8 - day | N/A | | | | |
| **Latent Heat** | 500 m | 8 - day | N/A | | | | |




**Table 2: Catchment properties with percentages of soil type and vegetation type, and the total number of pixels for different grid cell resolutions (at Demnitzer Millcreek Catchment outlet).**

| | | Spatial properties | | | |
|---|---|---|---|---|---|
| **Scale (m)** | | 250 | 500 | 750 | 1000 |
| **Number of Pixels** | | 1181 | 307 | 133 | 77 |
| **Soil Types (%)** | **Brown Earth** | 72.1 | 73.1 | 72.5 | 72.7 |
| | **Gley** | 5.0 | 4.8 | 4.8 | 5.2 |
| | **Podzol** | 14.4 | 13.9 | 14.2 | 13.9 |
| | **Peat** | 8.5 | 8.2 | 8.5 | 8.2 |
| **Vegetation Types (%)** | **Crops** | 51.9 | 52.2 | 52.6 | 51.8 |
| | **Conifers** | 29.2 | 29.3 | 28.9 | 29.6 |
| | **Broadleaf** | 6.0 | 5.8 | 6.0 | 6.1 |
| | **Pasture** | 12.9 | 12.7 | 12.5 | 12.5 |





**Table 3: Sensitive model parameters and descriptions for soil, vegetation and channel properties. A full list of EcH$_2$O model parameters can be found in Maneta and Silverman (2013).**

## Parameters

### Soil Properties

### Vegetation Properties

| *Description* | *Symbol* | *Name* | *Symbol* |
|---|---|---|---|
| Effective Brooks-Corey Parameter (-) | $\lambda_{BC}$ | Exponential parameter controlling vegetation vertical rooting distributions (-) | $K_{root}$ |
| Effective horizontal hydraulic conductivity (m/s) | $K_{eff}$ | Vegetation light attenuation coefficient (Beer's law) affecting the translation of radiation through the canopy | $K_{beers}$ |
| Effective soil air entry pressure (m) | $\Psi_{ae}$ | Maximum vegetation stomatal conductance under optimal conditions (m/s) | $g_{s,max}$ |
| Effective Soil Porosity (m3/m3) | $\varphi$ | Specific leaf water storage per leaf area index (m/LAI) | $CWS_{max}$ |
| Seepage to the channel | $Seep$ | Stomatal sensitivity to light (-) | $g_{s,light}$ |
| Snowmelt Coefficient (m/s*C) | $S_{melt}$ | Stomatal sensitivity to vapour pressure deficit (-) | $g_{s,vpd}$ |
| Soil Albedo (-) | $a_{soil}$ | **Channel Properties** | |
| Soil Depth (m) | $d$ | *Name* | *Symbol* |
| Vertical Leakance to Bedrock (-) | $L$ | Channel surface roughness | $R_{chan}$ |
| Vertical-Horizontal Anisotropy (-) | $K_vK_{eff}$ | Manning's n (in-stream channel roughness) | $Mn$ |





**Table 4: Median model efficiency from multi-criteria calibration (the brackets indicate the efficiency criteria). Superscripts**
**indicate a significant difference of efficiency between model scales, where a is 250 v. 500m, b is 250 v. 750m, c is 250 v. 1000m, d is**
**500 v. 750m, e is 500 v. 1000m, and f is 750 v 1000m.**

| Calibration to Field Data | | | | | | | | | | |
|---|---|---|---|---|---|---|---|---|---|---|
| | | Scale | | | | | Scale | | | |
| | | **250m** | **500m** | **750m** | **1000m** | | **250m** | **500m** | **750m** | **1000m** |
| **Discharge (NSE)** | **Demnitz Mill** | 0.69$^{abc}$ | 0.61$^e$ | 0.6 | 0.58 | | -1.01 | -0.86 | -0.81 | -0.81 |
| | **Demnitz** | 0.52$^{abc}$ | 0.44$^{de}$ | 0.35 | 0.34 | | -0.57 | -0.50 | -0.46 | -0.47 |
| **Stream Isotopes (NRMSE)** | **Peat North** | 0.01$^{abc}$ | 0.02$^e$ | 0.02$^f$ | 0.02 | | 2.28 | 2.53 | 2.51 | 2.76 |
| | **Peat South** | 0.02$^{abc}$ | 0.03$^d$ | 0.03 | 0.03 | | 2.79 | 3.03 | 3.26 | 3.14 |
| | **Bruchmill** | 0.02$^{abc}$ | 0.03$^{de}$ | 0.03$^f$ | 0.03 | | 2.42 | 2.66 | 2.70 | 2.76 |
| | **Demnitz Mill** | 0.03$^{abc}$ | 0.04$^{de}$ | 0.05 | 0.04 | | 2.72 | 2.85 | 3.13 | 3.05 |
| **Tr (NRMSE)** | **Forest** | 0.89$^{abc}$ | 1.02$^e$ | 1.05 | 1.07 | | 1.31 | 1.45 | 1.48 | 1.49 |
| **Soil Moisture (NSE, 20cm)** | **Cropland** | -0.06$^b$ | -0.35 | -0.58 | -0.71 | | -1.56 | -1.47 | -1.41 | -1.35 |
| | **Forest** | 0.43$^{ab}$ | -0.1$^e$ | -0.01$^f$ | 0.37 | | -1.78 | -1.59 | -1.84 | -1.91 |
| **Soil Isotopes (NRMSE, 20cm)** | **Forest** | 0.1$^{ac}$ | 0.18$^d$ | 0.1$^f$ | 0.15 | | 3.39 | 4.00 | 3.46 | 3.77 |
| **Groundwater Isotopes (NRMSE)** | **GW 4** | 0.05$^{abc}$ | 0.07 | 0.08 | 0.09 | | 2.77 | 2.81 | 2.77 | 2.84 |
| | **GW 8** | 0.05$^b$ | 0.05$^{de}$ | 0.03$^f$ | 0.05 | | 2.51 | 2.86 | 2.43 | 2.90 |
| Calibration to Remotely Sensed Data | | | | | | | | | | |
| | | **250m** | **500m** | **750m** | **1000m** | | **250m** | **500m** | **750m** | **1000m** |
| **Evapo-transpiration (NSE)** | **Forest** | 0.58$^{abc}$ | 0.5$^{de}$ | 0.66 | 0.67 | | 1.29 | 1.34 | 1.35 | 1.40 |
| | **Cropland** | 0.57 | 0.54 | 0.55 | 0.57 | | 1.23 | 1.38 | 1.39 | 1.37 |
| | **Conifers** | 0.07$^{abc}$ | 0.31 | 0.37 | 0.28 | | 1.56 | 1.57 | 1.54 | 1.44 |
| **Latent Heat (NSE)** | **Forest** | 0.34$^{bc}$ | 0.37$^{de}$ | 0.56 | 0.54 | | 4.70 | 4.75 | 4.72 | 4.78 |
| | **Cropland** | 0.3$^c$ | 0.28$^e$ | 0.3$^f$ | 0.38 | | 4.72 | 4.81 | 4.82 | 4.79 |
| | **Conifers** | 0.14$^{abc}$ | 0.35 | 0.4 | 0.34 | | 4.90 | 4.92 | 4.89 | 4.79 |
| **Soil Moisture (NSE)** | **Forest** | -0.28$^a$ | -1.04$^{de}$ | -0.57 | -0.68 | | -1.31 | -1.25 | -1.49 | -1.38 |
| | **Cropland** | 0.24$^{abc}$ | -0.13$^{de}$ | -0.58 | -0.83 | | -1.60 | -1.45 | -1.34 | -1.24 |

*Negative Loglikelihood* (vertical label for right half of table, both sections)





**Table 5: Mean catchment outflow contribution over the simulation period (2009 – 2019) as a proportion of the total catchment outflow, and the percent change of catchment storage from the beginning of 2009 to the end of 2019. Contributions include** 790 **evapotranspiration (ET), soil evaporation (Es), interception evaporation (Ei), transpiration (Tr), leakage, discharge, and groundwater outflow (GWout). Standard deviations of the contributions and storage changes are derived from the 100 'best' simulations for each resolution.**

| | | 250m | 500m | 750m | 1000m |
|---|---|---|---|---|---|
| **Outflow Contributions** | ET | 83.7 ± 4.9 % | 84.3 ± 4.8 % | 84.7 ± 9.6 % | 87.5 ± 6.7 % |
| | Es | 9.4 ± 2.7 % | 9.4 ± 3.0 % | 11.0 ± 3.7 % | 12.5 ± 4.4 % |
| | Ei | 24.4 ± 7.3 % | 25.3 ± 7.6 % | 21.7 ± 6.8 % | 22.7 ± 7.0 % |
| | Tr | 49.8 ± 7.9 % | 49.5 ± 8.6 % | 52.1 ± 9.8 % | 52.2 ± 8.2 % |
| | Leakage | 2.4 ± 3.9 % | 3.0 ± 6.0 % | 4.1 ± 8.5 % | 1.9 ± 4.8 % |
| | Discharge | 13.9 ± 3.5 % | 12.8 ± 5.0 % | 11.1 ± 4.8 % | 10.6 ± 5.2 % |
| | GWout | 0.0 ± 0.0 % | 0.0 ± 0.0 % | 0.0 ± 0.0 % | 0.0 ± 0.1 % |
| **Change in Storage** | Soil Layer 1 | -0.8 ± -1.1 % | -0.7 ± -1 % | -0.4 ± -1.0 % | -0.7 ± -1.0 % |
| | Soil Layer 2 | -3.0 ± -3.2 % | -2.8 ± -2.7 % | -2.0 ± -3.0 % | -3.7 ± -2.6 % |
| | Soil Layer 3 | -15.0 ± -17.4 % | -19 ± -28 % | -19.4 ± -21.6 % | -18.0 ± -18.9 % |
| | Groundwater | -81.2 ± -40.6 % | -77.5 ± -53.8 % | -78.2 ± -64.9 % | -77.7 ± -59.7 % |





**Table 6: Estimated stream water ages (years) under high flow anomaly (Qa>1.0), normal flows (-0.5≤ Qa≤1.0), and below average flow anomaly (Qa≤-0.5) for Peat North, Peat South, Bruchmill, and Demnitz Mill.**

| | | 250m | 500m | 750m | 1000m |
|---|---|---|---|---|---|
| **Peat North** | $Q_a < -0.5$ | $6.9 \pm 0.8$ | $12.9 \pm 1.9$ | $12.3 \pm 2$ | $16.2 \pm 3.1$ |
| | $-0.5 \leq Q_a \leq 1.0$ | $6.6 \pm 1.2$ | $13.4 \pm 2.6$ | $12.7 \pm 2.6$ | $16.2 \pm 3.6$ |
| | $Q_a > 1.0$ | $6.1 \pm 1.1$ | $13.1 \pm 2.3$ | $12.1 \pm 2.4$ | $14.0 \pm 3.3$ |
| **Peat South** | $Q_a < -0.5$ | $6.9 \pm 0.8$ | $12.7 \pm 2.1$ | $12.3 \pm 2.4$ | $15.5 \pm 3.8$ |
| | $-0.5 \leq Q_a \leq 1.0$ | $6.6 \pm 1.2$ | $13.0 \pm 2.8$ | $12.4 \pm 3.0$ | $15.0 \pm 3.7$ |
| | $Q_a > 1.0$ | $6.1 \pm 1.1$ | $11.6 \pm 2.6$ | $11.0 \pm 2.7$ | $11.4 \pm 3.0$ |
| **Bruch-mill** | $Q_a < -0.5$ | $7.5 \pm 1.0$ | $12.3 \pm 2.2$ | $11.8 \pm 2.3$ | $14.8 \pm 3.6$ |
| | $-0.5 \leq Q_a \leq 1.0$ | $7.4 \pm 1.3$ | $12.7 \pm 2.9$ | $12.1 \pm 2.9$ | $14.6 \pm 3.7$ |
| | $Q_a > 1.0$ | $6.6 \pm 1.2$ | $11.2 \pm 2.6$ | $10.2 \pm 2.5$ | $11.1 \pm 3.1$ |
| **Demnitz Mill** | $Q_a < -0.5$ | $7.5 \pm 1.0$ | $12.1 \pm 2.2$ | $11.7 \pm 2.3$ | $14.6 \pm 3.6$ |
| | $-0.5 \leq Q_a \leq 1.0$ | $7.4 \pm 1.3$ | $12.5 \pm 2.9$ | $12.1 \pm 2.9$ | $14.6 \pm 3.8$ |
| | $Q_a > 1.0$ | $6.6 \pm 1.2$ | $10.9 \pm 2.7$ | $10.1 \pm 2.5$ | $11.2 \pm 3.2$ |