# Peer review of "Quantifying the effects of land-use and model scale on water partitioning and water ages using tracer-aided ecohydrological models"

_Hydrology and Earth System Sciences, 2020_

## Referee Comment (RC1) · Anonymous Referee #1 · 27 Nov 2020

General comments

The paper presents substantial new results on the effects of model resolution on eco-hydrolgical flux simulation using supplementary isotope data. The overall quality of the paper is very good, but some key aspects of the methodology are unclear in the main manuscript, and are only outlined in the supplementary material. Other than a few details, the paper is clear and well supported. I see no reason not to accept the manuscript for publication in HESS, subject to a few minor corrections.

Specific comments

74-80: The research questions are clear enough but three fairly complex questions

seems excessive for a single research paper. Having read through the manuscript, all the questions are addressed to some degree, but they are an unfocused introduction to the aim and scope of the presented research.

86: Describing a 66 km2 catchment as 'mesoscale' is dubious; it falls short even of your previous definition of 100km2. Let the area speak for itself, as was done in the abstract.

210: Is 'climate zone' the best word choice? It doesn't necessarily bring weather stations and Thiessen polygons to mind.

233: It is stated that NRMSE is used, but not what is used to normalize the error. The justification for using the NRMSE is also insufficient to explain why the NSE was rejected (the NSE is fundamentally a normalized squared error) for the isotope simulations. Either a better justification in text or a reference is needed.

258-260: The multi-criteria calibration methodology is a key part of this research, but it would not be possible to replicate the method with the description provided here. Too many significant details on the method have been relegated to the supplement.

374: Unless there is some kind of character limit on the manuscript, the readability might be improved by using actual process terms rather than contractions. Tr and Re were defined but in a different section.

Technical corrections:

66: Squared kilometers is not properly superscripted. This occurs irregularly throughout the manuscript.

81: Likely grammatical error, unsure what the subject of 'sought' is.

207: Is there too many 'of' in the length description?

236: Supplementary material referenced by letter, but the supplement is numbered. Not the only location this occurs.

[Figure]

Figure 1: There is no scale on the main map in (a)

[Figure]

---

## Referee Comment (RC2) · Anonymous Referee #2 · 23 Dec 2020

Summary:

This paper evaluates the performance of an ecohydrological model equipped with water isotope module (EcH2O-iso)in simulating soil water, evapotranspiration, recharge or runoff, groundwater, and water ages across different spatial scales (250, 500, 750, and 1000m). Moreover, the study addresses three main research questions that will be revealed using the model, applied in an intensively monitored catchment in northeast Germany. They conclude that the model with a coarser resolution (1000m) was unable to replicate the observed streamflow and distributed isotope dynamics and simulates higher evapotranspiration, lower relative transpiration, increased overland flow, and

slower groundwater movement. In addition, they also conclude that water isotopes provide effective calibration constraints for larger resolution model and help understand the influence of grid-resolution on the simulation of vegetation-soil interactions.

Assessment:

The topic of the paper fits well in HESS and is of great interest for the HESS readers. The findings are also interesting for readers working on the ecohydrological model development, land-surface fluxes quantification, and the use of water isotopes in hydrological applications. The paper at the present state still needs some revisions. The authors need to elaborate more on the method section and the conclusion does not summarize all three research questions brought up in this paper. I provide my general and detailed comments below and would ask the authors to take these comments into consideration as they revise the paper.

General comment: L refers to Line and P refers to Page.

1. In my opinion, the title is not mirroring the aim of this paper. The title is: upscaling land-use effects on water partitioning and water ages using tracer-aided ecohydrological model (L1) while the aim of this paper is to explore the changes in the skill of an ecohydrological model in capturing flux, storage, and mixing dynamics across spatial scales (L65). When I read the paper thoughtfully, the paper discusses the performance of the model to simulate the water fluxes and ages across different spatial scales. Therefore, I suggest to change the title.

2. The study tries to answer three research questions (L74-80) and they are explained in the discussion section. For the conclusion section, however, a summary of the answers for these research questions is not clearly presented. I could not find a conclusion for research question two or maybe I miss it.

3. The EcH2O-iso model consists of three main parts, which are the energy balance model, the water balance model, and the water isotope module. I understand that the

idea of the paper is not to discuss model development in a detailed manner as it was discussed in the previous studies (e.g., Maneta and Silverman, 2013 and Kuppel et al., 2018). However, there are not so many hydrological models equipped with water isotope module compared to the climate model. I expected that the authors would provide more information about how isotopic mixing and fractionation are performed in the model. Please elaborate more about the mixing and fractionation in all water components, such as in precipitation, transpiration, soil, storage, and GW. Also what methods do the authors use to calculate isotopic fractionation in different fluxes such e.g. the Craig-Gordon method, Keeling plot, or a steady-state method introduced by Dongmann et al. (1974) to calculate the isotopic composition of leaf water? How to define the isotopic composition of river water since it is mixing between groundwater, precipitation, and river from the upstream. When river water flowing in the channel, does it undergo evaporation fractionation, or neglected?

4. In the discussion section, the authors found that for forests, the evapotranspiration is higher, the recharge is lower, and the transpiration fraction is lower compared to croplands (L407-409). They argue that the lower Tr ratio is due to the interception. Could the authors elaborate more on why and what are the possible reasons? Also please provide all the evapotranspiration fractions here (soil evaporation, transpiration, and interception for crops and forest). I found in many studies that taller plants transpire more water than shorter plants (e.g., Oaks vs. wheat, Xu et al., 2008; Zhang et al., 2011, cotton field vs. corn and soybean, Kool et al., 2014). For some field measurements, the transpiration fraction for forests is 65-76% and 60% for grass (Choudhury and DiGirolamo, 1998; Blyth and Harding, 2011).

Line by line comment:

L1: for the abstract, please see the general comment.

L15: is it not better if the authors write "different scales" instead of only "scale"?

L65: here the readers can see the aim of the study, which differs from the title (upscaling land-use effects instead of evaluating the model skill in simulating different water fluxes across different spatial scales).

L76-77: the second question is not clearly answered in the conclusion.

L86-87: the authors may rephrase the sentence into: "The 66 km2 Demnitzer Millcreek Catchment (DMC) located 55 km east of Berlin (52°23'N, 14°14'E) is a mesoscale catchment that receives 575 mm of precipitation annually"

L87: the authors may change the words: "(372 to 776 mm/year)" into "from 372 to 776 mm/year".

L88: I think the general term is "convective" storms and not convectional storms.

L94: missing comma: ...unavailable, these were......

L95: first, ERA 5 is not a remote sensing dataset. It is re-analysis product that is fussed with all types of observations (not only remote sensing). Second, it is more common and general to use the word "sensing" instead of "sensed". I cannot provide a good argument about it but you may look at: https://www.researchgate.net/post/Remote-sensing-data-or-images-vs-remotely-sensed-data-or-images-Are-they-both-OK

L131: the authors may revise the text into: "......during periods of streamflow measurements". For the next sentence, the authors may revise the sentence to avoid the use of the words "periods of streamflow" two times.

L132: the authors may revise the text into: "Evaporation was prevented by applying a layer of......"

L134: missing "were" analysed. What is direct-equilibrium method?

L135: the authors may replace the word "with" with "using"

L141: what is "cf"?

L145: what kind of components? It is an unclear word.

L146: I am just wondering if the model was designed to be forced with RCM only or can it be forced with gridded in situ observations? This study also used in situ observations and not RCM.

L149: I suggest to replace ";" with full stop "."

L172: Could the authors indicate layers 1 to 3 in Fig. S1? It will help the readers to discriminate layers.

L197: missing space.

L200: please provide a reference for the n values.

L210: they are locations and not climate zones. Also please provide the names for these five locations in the main text.

L221: How about streamflow isotopic composition?

L224: I am wondering why the use of NSE and NRMSE is inconsistent. Some variables were evaluated using NSE and some using NRMSE.

L247: I am wondering what are the variable constraints do the authors use to calibrate the model? Is it only discharge or the authors consider all variables such as GW, SM.

L260: Fig. S3 is for validation and not for calibration.

L269: this is correct: ERA5 reanalysis products.

L269-271: I am a bit confused here. How do the authors calibrate the isotopes?

L276: the authors may replace the word "very few parameters" with "a few parameters"

L277: Do I miss supplementary material B? I could not find it.

L279: two commas there are not needed.

L284: How about streamflow?

L285-287: here the authors mention soil moisture in layers 1 and 2, however, Figure 2c does not distinguish between layers 1 and 2. Where can I look at the results?

L295: I suggest to add the word "that" in between processes and were.

L311: I am wondering why the Ei is somehow twice as higher as soil evaporation and half of Tr. If I look at the land cover map (Fig. 1a), I see the land cover, in general, can be divided into half forest and half arable land. The Ei is indeed higher for the forest but it cannot exceed the TR, and EI is very low or even insignificant for arable land. What are the reasons? Do the authors think this is the general problem (underestimation of Tr) found in many models as it was discussed by Sutanto et al., 2014?

L316: If the discharge is discussed first, why do not the authors swap the Figure between e and f with a and b? Hence, Figure 3a and B will be for streamflow results.

L324: the authors may revise the sentence into "......flow events that are not present......"

L329: It is not the correct sub-title.

L330: here the authors only mention Tr. How about the Es and Ei?

L336: I am just wondering why the authors do not use the blue color for positive and red color for negative correlation (inverse colors)? Usually red represents low value and blue represent high value.

L337: please provide the value for the fraction of Tr.

L343: from 38 mm/year to 22 mm/year is not a slight decrease, it is almost half.

L39-350: the authors claim that the decrease in annual recharge is largely linked to ET, which is mainly from high Ei. However, I cannot see the Ei results in the suggested Figures (Fig.1, Fig.4, and Fig. S5) or even in any figure.

L363: Again, here I could not see Supplementary Material B.

L370: Can the authors explain why transpiration age is longer than soil in layers 1 and 2 and in GW.

L371: I am a bit confused here. I see that it is longer and not lower. e.g. at 250 model resolution, it is 47 days while at 1000 model resolution it is 28 days.

L381: Remove Figure S5. I do not see the results in Fig. S5. Fig. S5 is for the next sentence.

L384-385: here the authors discuss the GW age, however, I could not see the results. Is it in Figure S5? Do the authors mean GW age as L3? I only see GW storage.

L393-395: I could not see the results for stream water age of 0.5 and 1.8 years during large events in Table 6.

L402: In my opinion, it is not minor variability.

L407-409: see my general comment point 4.

L467: it is "are" and not "is"

L482: remove the second is in between coupling and with.

L486: the authors may revise the sentence into: "......scales, long-term and multi-scale data collection are......"

L489-490: the authors may revise the text in the brackets as (e.g. spatial resolution for isotopes and temporal resolution for sapflow).

L498: in my opinion, I will limit to 500 m maximum due to the highest uncertainty in model results above 500 m (Figure 3, 5).

L500: Figures 6e-h are not the correct figures.

L516: the authors may remove the words "have been identified"

L526-527: missing "that": ......fluxes that are additionally expected to change is not

well known.

P26: Figure 3. Swap a, b for discharge first. To increase figure readability, the map can be removed since it is already available in Fig. 1a.

P28: Figure 5. I could not see the measured SM (dashed line).

P31: Table 1. Suggestion: please indicate in the table which one is obtained from ERA5 and which one is from MODIS. e.g. instead of location N/A, why not write ERA5 or MODIS? Also please keep in mind ERA 5 is not categorized as remote sensing data.

P32: missing Table 2 borderline.

P34: Table 4. What is negative Loglikelihood? It is not explained in the text.

P35: Table 5. Please mention that Es, Ei, and Tr were partitioned from ET.

References:

1. Maneta, M. P., and Silverman, N. L.: A Spatially Distributed Model to Simulate Water, Energy, and Vegetation Dynamics Using information from Regional Climate Models, Earth Interactions, 17, 1-44, doi:10.1175/2012ei000472.1, 2013.

2. Kuppel, S., Tetzlaff, D., Maneta, M. P., and Soulsby, C.: EcH2O-iso 1.0: water isotopes and age tracking in a process-based, distributed ecohydrological model, Geoscientific Model Development, 11, 3045-3069, doi:10.5194/gmd-11-3045-2018, 2018.

3. Dongmann, G., Nurnberg, H. W., Forstel, H., and Wagener, K.: On the enrichment of $H_2^{18}O$ in leaves of transpiring plants, Radiat. Environ. Bioph., 11, 41–52, 1974.

4. Xu, Z., Yang, H., Liu, F., An, S., Cui, J., Wang, Z., and Liu, S.: Partitioning evapotranspiration flux components in a subalpine shrubland based on stable isotopic measurements, Bot. Stud., 49, 351-361, 2008.

5. Zhang, Y., Shen, Y., Sun, H., and Gates, J. B.: Evapotranspiration and its partitioning in an irrigated winter wheat field: A combined isotopic and micrometeorologic

approach, J. Hydrol., 408, 203– 211, 2011.

6. Kool, D., Agam, N., Lazarovitch, N., Heitman, J. L., Sauer, T. J., and Ben-Gal, A.: A review of approaches for evapotranspiration partitioning, Agr. Forest Meteorol., 184, 56-70, doi:10.1016/j.agrformet.2013.09.003, 2014.

7. Choudhury, B. and DiGirolamo, N.: A biophysical process-based estimate of global land surface evaporation using satellite and ancillary data-I. Model description and comparison with observations, J. Hydrol., 205, 164–185, 1998.

8. Blyth, E. M. and Harding, R. J.: Methods to separate observed global evapotranspiration into the interception, transpiration and soil surface evaporation components, Hydrol. Process., 25, 4063–4068, doi:10.1002/hyp.8409, 2011.

9. Sutanto, S. J., van der Hurk, B., Dirmeyer, P. A., Seneviratne, S. I., Röckmann, T., Trenberth, K. E., Blyth, E. M., Wenninger, J., and Hoffmann, G.: HESS Opinions "A perspective on isotope versus non-isotope approaches to determine the contribution of transpiration to total evaporation, Hydrol. Earth Syst. Sci., 18, 2815–2827, doi:10.5194/hess-18-2815-2014, 2014.

---

## Author Comment (AC1) · 11 Jan 2021

General Comments: The paper presents substantial new results on the effects of model resolution on ecohydrological flux simulation using supplementary isotope data. The overall quality of the paper is very good, but some key aspects of the methodology are unclear in the main manuscript, and are only outlined in the supplementary material. Other than a few details, the paper is clear and well supported. I see no reason not to accept the manuscript for publication in HESS, subject to a few minor corrections.

Response to General Comments: The authors thank the reviewer for their comments. We will clarify the methodology presented in the manuscript to reduce dependencies

on the supplementary material.

Specific Comments

R1C1: 74-80: The research questions are clear enough but three fairly complex questions seems excessive for a single research paper. Having read through the manuscript, all the questions are addressed to some degree, but they are an unfocused introduction to the aim and scope of the presented research.

Response to R1C1: The authors thank the reviewer for their suggestion. Through the revision, the authors will refocus the research questions to better present the aims and scope of the research conducted.

R1C2: 86: Describing a 66 km2 catchment as 'mesoscale' is dubious; it falls short even of your previous definition of 100km2. Let the area speak for itself, as was done in the abstract.

Response to R1C2: The authors will clarify 'mesoscale' as a descriptor in the manuscript. We wanted to differentiate the site as being larger than a small experimental catchment (typically <10km2 and often <1km2).

R1C3: 210: Is 'climate zone' the best word choice? It doesn't necessarily bring weather stations and Thiessen polygons to mind.

Response to R1C3: The authors thank the reviewer for their suggestion. The authors will revise the wording to "...forcing data were included as representative polygon areas from five local climate stations".

R1C4: 233: It is stated that NRMSE is used, but not what is used to normalize the error. The justification for using the NRMSE is also insufficient to explain why the NSE was rejected (the NSE is fundamentally a normalized squared error) for the isotope simulations. Either a better justification in text or a reference is needed.

Response to R1C4: The authors apologize for the error in the manuscript. The RMSE

and NRMSE were exclusively used for the sensitivity analysis. Calibration efficiency was measured by NSE and normalized mean absolute error (NMAE). The authors agree that there is limited usefulness of NRMSE if NSE is used. The NMAE was used because the authors did not want to over-emphasis peak values when data collection had inconsistent time-steps and the isotope response was quite subdued. The authors will revise the description here to clarify the use of MAE, not RMSE.

R1C5: 258-260: The multi-criteria calibration methodology is a key part of this research, but it would not be possible to replicate the method with the description provided here. Too many significant details on the method have been relegated to the supplement.

Response to R1C5: The authors thank the reviewer for their suggestion. To keep the manuscript streamlined, the authors will move the most significant details of the multi-criteria calibration from the supplementary material to the manuscript. However, we are mindful of maintaining a balance between detail and excessive manuscript length and using the supplementary material to achieve this.

R1C6: 374: Unless there is some kind of character limit on the manuscript, the readability might be improved by using actual process terms rather than contractions. Tr and Re were defined but in a different section.

Response to R1C6: To aid with readability and maintain consistency, the authors will update the Tr and Re terms throughout the manuscript to only use the actual process terms.

R1C7: 66: Squared kilometres is not properly superscripted. This occurs irregularly throughout the manuscript.

Response to R1C7: Through the revision, the authors will check for proper sub- and superscripts and update accordingly.

R1C8: 81: Likely grammatical error, unsure what the subject of 'sought' is.

Response to R1C8: The authors will revise the wording to improve clarity.

R1C9: 207: Is there too many 'of' in the length description?

Response to R1C9: The authors will remove the erroneous extra words in the description.

R1C10: 236: Supplementary material referenced by letter, but the supplement is numbered. Not the only location this occurs.

Response to R1C10: The authors apologize for this confusion. The supplementary material lettering will be updated to numbers to be consistent with the appended supplementary material.

R1C11: Figure 1: There is no scale on the main map in (a)

Response to R1C11: The authors will add a scale to the main map of Figure 1a

---

## Author Comment (AC2) · 11 Jan 2021

General Comments: This paper evaluates the performance of an ecohydrological model equipped with water isotope module (EcH2O-iso)in simulating soil water, evapotranspiration, recharge or runoff, groundwater, and water ages across different spatial scales (250, 500, 750, and 1000m). Moreover, the study addresses three main research questions that will be revealed using the model, applied in an intensively monitored catchment in northeast Germany. They conclude that the model with a coarser resolution (1000m) was unable to replicate the observed streamflow and distributed isotope dynamics and simulates higher evapotranspiration, lower relative transpiration,

increased overland flow, and slower groundwater movement. In addition, they also conclude that water isotopes provide effective calibration constraints for larger resolution model and help understand the influence of grid-resolution on the simulation of vegetation-soil interactions.

Assessment: The topic of the paper fits well in HESS and is of great interest for the HESS readers. The findings are also interesting for readers working on the ecohydrological model development, land-surface fluxes quantification, and the use of water isotopes in hydrological applications. The paper at the present state still needs some revisions. The authors need to elaborate more on the method section and the conclusion does not summarize all three research questions brought up in this paper. I provide my general and detailed comments below and would ask the authors to take these comments into consideration as they revise the paper. Response to General Comments The authors thank the reviewer for their comments on this manuscript, which have aided in adding clarity in presenting model results and objectives.

General Comment 1: In my opinion, the title is not mirroring the aim of this paper. The title is: upscaling land-use effects on water partitioning and water ages using traceraided ecohydrological model (L1) while the aim of this paper is to explore the changes in the skill of an ecohydrological model in capturing flux, storage, and mixing dynamics across spatial scales (L65). When I read the paper thoughtfully, the paper discusses the performance of the model to simulate the water fluxes and ages across different spatial scales. Therefore, I suggest to change the title.

Response to General Comment 1: The authors thank the reviewer for their insights. The authors will revise the title of the manuscript to better reflect the main aims as suggested. The authors will revise the title to "Quantifying the effects of land-use and model scale on water partitioning and water ages using tracer-aided ecohydrological models"

General Comment 2: The study tries to answer three research questions (L74-80) and

they are explained in the discussion section. For the conclusion section, however, a summary of the answers for these research questions is not clearly presented. I could not find a conclusion for research question two or maybe I miss it.

Response to General Comment 2: The authors apologize for not clearly presenting the second research question in the conclusions. During revision, the authors will clearly state the summary of the main findings on the effect of model resolution on the robustness of the model in the abstract and the conclusions.

General Comment 3: The EcH2O-iso model consists of three main parts, which are the energy balance model, the water balance model, and the water isotope module. I understand that the idea of the paper is not to discuss model development in a detailed manner as it was discussed in the previous studies (e.g., Maneta and Silverman, 2013 and Kuppel et al., 2018). However, there are not so many hydrological models equipped with water isotope module compared to the climate model. I expected that the authors would provide more information about how isotopic mixing and fractionation are performed in the model. Please elaborate more about the mixing and fractionation in all water components, such as in precipitation, transpiration, soil, storage, and GW. Also what methods do the authors use to calculate isotopic fractionation in different fluxes such e.g. the Craig-Gordon method, Keeling plot, or a steady-state method introduced by Dongmann et al. (1974) to calculate the isotopic composition of leaf water? How to define the isotopic composition of river water since it is mixing between groundwater, precipitation, and river from the upstream. When river water flowing in the channel, does it undergo evaporation fractionation, or neglected?

Response to General Comment 3: The authors thank the reviewer for their suggestion to further elaborate on the isotope and water age mixing section. The authors recognize that isotope mixing modules are not present in many models. However, the authors believe that repeated explanation of the same mixing process for each storage will be quite repetitive. The authors will clarify how mixing is conducted (amount weighted average of inputs and storage) and how transpiration mixing is conducted.

The authors will state that the Craig-Gordon model is used to estimate the fractionation in the soils and in the open water. The manuscript currently states that open water channel evaporation is subject to fractionation (P7L200).

Response to General Comment 4: In the discussion section, the authors found that for forests, the evapotranspiration is higher, the recharge is lower, and the transpiration fraction is lower compared to croplands (L407-409). They argue that the lower Tr ratio is due to the interception. Could the authors elaborate more on why and what are the possible reasons? Also please provide all the evapotranspiration fractions here (soil evaporation, transpiration, and interception for crops and forest). I found in many studies that taller plants transpire more water than shorter plants (e.g., Oaks vs. wheat, Xu et al., 2008; Zhang et al., 2011, cotton field vs. corn and soybean, Kool et al., 2014). For some field measurements, the transpiration fraction for forests is 65-76% and 60% for grass (Choudhury and DiGirolamo, 1998; Blyth and Harding, 2011).

Response to General Comment 4: There are numerous factors that, in this catchment, result in a somewhat unexpected ratio of transpiration to ET in the forested areas. Firstly, it should be noted that the values presented in the manuscript are annual average fractions, which are moderately skewed toward lower transpiration to ET ratios due to winter interception and soil evaporation. This will be particularly marked in the conifer forests where the LAI remains constant year-round facilitating higher interception storage and evaporation. On this point, the authors will add clarification in the discussion on the seasonality of the transpiration to ET ratio influence in the conifer forests. To further clarify, the authors will include a time series of the average evapotranspiration components for the dominant vegetation types to show this variability. Secondly, while the transpiration to ET ratio is lower in the conifer forests, the dominant soil type in the forests is also different from the croplands, being less water retentive. This soil type limits the water available for the trees, introducing water stress and reducing transpiration further from potential (not energy limited). The observed transpiration to ET ratio in the study is similar to those given by Choudhury and DiGirolamo, 1998 and Blyth

and Harding, 2011. To further clarify the reason for the low transpiration due to dry soil conditions, the authors will add to the discussion clarifying the water-limited conditions.

Specific Comments

R2C1: L1: for the abstract, please see the general comment.

Response to R2C1: The authors will revise the abstract as suggested in the general comments.

R2C2: L15: is it not better if the authors write "different scales" instead of only "scale"?

Response to R2C2: The authors will revise the wording.

R2C3: L65: here the readers can see the aim of the study, which differs from the title (upscaling land-use effects instead of evaluating the model skill in simulating different water fluxes across different spatial scales).

Response to R2C3: The authors will clarify on revision.

R2C4: L76-77: the second question is not clearly answered in the conclusion.

Response to R2C4: The authors will add the statement "Identification of sensitive parameters was similar across scales; however, a notable decrease in the degree of sensitivity, coupled with an increase in all model output uncertainty, occurred with coarser model resolutions." to the conclusions to directly address the primary findings of the second research question.

R2C5: L86-87: the authors may rephrase the sentence into: "The 66 km2 Demnitzer Millcreek Catchment (DMC) located 55 km east of Berlin (52_23'N, 14_14'E) is a mesoscale catchment that receives 575 mm of precipitation annually"

Response to R2C5: The authors will revise the wording of the sentence.

R2C6:L87: the authors may change the words: "(372 to 776 mm/year)" into "from 372 to 776 mm/year".

Response to R2C6: The authors will change the wording as suggested by the reviewer.

R2C7: L88: I think the general term is "convective" storms and not convectional storms.

Response to R2C7: The authors will change the terminology presented in revision.

R2C8: L94: missing comma . . .unavailable, these were . . .

Response to R2C8: The authors will revise the grammar of the sentence.

R2C9: L95: first, ERA 5 is not a remote sensing dataset. It is re-analysis product that is fussed with all types of observations (not only remote sensing). Second, it is more common and general to use the word "sensing" instead of "sensed". I cannot provide a good argument about it but you may look at: https://www.researchgate.net/post/Remotesensing-data-or-images-vs-remotely-sensed-data-or-images-Are-they-both-OK

Response to R2C9: The authors thank the reviewer for their suggestion. The authors will change the description of the ERA5 dataset to re-analysis data, and change instances of "sensed" to "sensing"

R2C10: L131: the authors may revise the text into: ". . .during periods of streamflow measurements". For the next sentence, the authors may revise the sentence to avoid the use of the words "periods of streamflow" two times.

Response to R2C10: To improve clarity, the authors will remove "during periods of streamflow" for both sentences, and will add a sentence stating "Isotopic samples of stream water were only taken when streams were flowing and not with standing water."

R2C11: L132: the authors may revise the text into: "Evaporation was prevented by applying a layer of

Response to R2C11: The authors will revise the text as suggested by the reviewer.

R2C12: L134: missing "were" analysed. What is direct-equilibrium method?

Response to R2C12: The authors will revise the wording of the sentence. The authors feel that the full description of the direct-equilibrium method is not necessary here and will make it clearer that the description of these data is given in Kleine et al., 2020.

R2C13: L135: the authors may replace the word "with" with "using"

Response to R2C13: The authors will replace "with" with "using".

R2C14: L141: what is "cf"?

Response to R2C14: cf is the APA Latin abbreviation of conferatur, meaning: comparable to/to make a comparison.

R2C15: L145: what kind of components? It is an unclear word.

Response to R2C15: The authors will revise the statement to "...integrates modules for soil and vegetation to simulate..."

R2C16: L146: I am just wondering if the model was designed to be forced with RCM only or can it be forced with gridded in situ observations? This study also used in situ observations and not RCM.

Response to R2C16: The authors thank the reviewer for the suggestion. The authors will revise the statement to indicate that the model was developed to use a variety of input forcing, from local climate stations to RCMs.

R2C17: L149: I suggest to replace ";" with full stop "."

Response to R2C17: The authors will make the suggested change.

R2C18: L172: Could the authors indicate layers 1 to 3 in Fig. S1? It will help the readers to discriminate layers.

Response to R2C18: The addition of layers 1, 2 and 3 would be relatively arbitrary (e.g. water table may vary between layers 1 and 3), and would further complication of the plot due to additional flux arrows to and from each soil layer.

R2C19: L197: missing space.

Response to R2C19: The authors will correct this error.

R2C20: L200: please provide a reference for the n values.

Response to R2C20: The earlier reference for the n values (Mathieu and Bariac, 1996; Braud et al., 2005) will be provided with this sentence to improve clarity.

R2C21: L210: they are locations and not climate zones. Also please provide the names for these five locations in the main text.

Response to R2C21: The authors will revise the text to indicate that these are 5 climate station, rather than climate zones. As the names of the stations do not have meaning to those outside of the area, the authors believe that the inclusion of the station names is not necessary.

R2C22: L221: How about streamflow isotopic composition?

Response to R2C22: The authors thank the reviewer for their comment. The authors will add that stream isotopic composition was initialized with measured stream isotopes in 2018 and 2019.

R2C23:L224: I am wondering why the use of NSE and NRMSE is inconsistent. Some variables were evaluated using NSE and some using NRMSE.

Response to R2C23: The authors apologize for the confusion regarding the efficiency criteria. The RMSE was only used for the sensitivity analysis, while NSE and normalized mean absolute error (NMAE) were used for calibration. MAE was used due to the inconsistent time-series and to reduce over-emphasis of peak values which may be present in time-series with longer periods between samples. The authors will make this correction during revision.

R2C24: L247: I am wondering what are the variable constraints do the authors use to calibrate the model? Is it only discharge or the authors consider all variables such as

[Figure]

GW, SM.

Response to R2C24: The authors thank the reviewer for their comment. The authors will move some of the calibration descriptions from the supplementary material to the main text to clarify the calibration method. The authors will provide a link to Table 1 to directly show the datasets used in calibration and validation.

R2C25 L260: Fig. S3 is for validation and not for calibration.

Response to R2C25: As stated in the caption of Fig. S3 in the supplementary material, the figure shows the ranges of the parameters from calibration.

R2C26: L269-271: I am a bit confused here. How do the authors calibrate the isotopes?

Response to R2C26: The author apologize for the confusion in the calibration section. Since isotopes were only available during the calibration period, the authors split some of the isotope data to use some during validation. The isotopes (stream, groundwater, and forest layer 1 isotopes) were calibrated with NMAE, and forest layer 2 and cropland layer 1 were used for validation. The authors will clarify the data used for calibration and validation in Table 1.

R2C27: L276: the authors may replace the word "very few parameters" with "a few parameters"

Response to R2C27: The authors will change the wording to "...is sensitive to few parameters..."

R2C28: L277: Do I miss supplementary material B? I could not find it.

Response to R2C28: Supplementary Material section 2 was erroneously labelled as Supplementary Material B. The authors will correct this during revision.

R2C29: L279: two commas there are not needed.

Response to R2C29: The authors will correct this during revision.

R2C30: L284: How about streamflow?

Response to R2C30: During revision, the authors will clarify that the "runoff generation" described in the sentence refers to the streamflow/discharge in Figure 2.

R2C31: L285-287: here the authors mention soil moisture in layers 1 and 2, however, Figure 2c does not distinguish between layers 1 and 2. Where can I look at the results?

Response to R2C31: Figure 2c shows both layers 1 and 2 (shown with headers SWCL1 and SWCL2, respectively). The authors recognize that the headers on the figure were not clear. During the revision, the authors will change the header description and caption to ease interpretation.

R2C32: L295: I suggest to add the word "that" in between processes and were.

Response to R2C32: The authors will revise the sentence to "...suggest that dominant catchment processes were..."

R2C33: L311: I am wondering why the Ei is somehow twice as higher as soil evaporation and half of Tr. If I look at the land cover map (Fig. 1a), I see the land cover, in general, can be divided into half forest and half arable land. The Ei is indeed higher for the forest but it cannot exceed the TR, and EI is very low or even insignificant for arable land. What are the reasons? Do the authors think this is the general problem (underestimation of Tr) found in many models as it was discussed by Sutanto et al., 2014?

Response to R2C33: The figures show the annual average components of Tr and Es from ET. Due to the more constant winter precipitation feeding interception, overall transpiration appears lower, particularly in the forests where sandier soils dominate and higher winter groundwater recharge limits the water available for spring/summer forest transpiration. Lower LAI in the arable lands decreases the potential interception evaporation relative to the forest. The sandier soils in the forest additional result

in lower soil moisture, resulting in soil moisture limited transpiration flux. For the ratio between interception evaporation and transpiration, the lower atmospheric resistance of interception evaporation results in lower transpiration fractions during extended periods of higher rainfall (i.e. more intercepted water). Additionally, the fractions presented here are consistent with those in Europe, with grasslands/maize (Sutanto et al., 2012; Herbst et al., 1996) showing values of 77-97% transpiration, consistent with the finding in the arable land in our study and a catchment average of 72-77% Tr/ET (Fig. 4). The authors will add this explanation to the discussion during the revision.

R2C34: L316: If the discharge is discussed first, why do not the authors swap the Figure between e and f with a and b? Hence, Figure 3a and B will be for streamflow results.

Response to R2C34: During revision, the authors will modify Figure 3 to show the discharge first (Fig 3a and b) and remove the map to aid with readability.

R2C35: L324: the authors may revise the sentence into "...flow events that are not present..."

Response to R2C35: The authors will revise the sentence as suggested by the reviewer.

R2C36: L329: It is not the correct sub-title.

Response to R2C36: The authors thank the reviewer for identifying this error. The authors will revise the sub-title during revision.

R2C37: L330: here the authors only mention Tr. How about the Es and Ei?

Response to R2C37: The authors only mention ET and Tr as they were the time series used in calibration. Es and Ei were not used in calibration and therefore are not relevant to include here. As with Response to R2C24, the authors will revise the calibration section to help clarify the data used for calibration.

R2C38: L336: I am just wondering why the authors do not use the blue color for positive and red color for negative correlation (inverse colors)? Usually red represents low value and blue represent high value.

Response to R2C38: The authors will reverse the colour scheme during the revision to show blue to positive correlation and red for the negative correlation.

R2C39: L337: please provide the value for the fraction of Tr.

Response to R2C39: The authors will provide a reference to Table 5 and Fig.4 which both show the annual average fraction of Tr (Table 5 with Tr as a fraction of total catchment loss).

R2C40 L343: from 38 mm/year to 22 mm/year is not a slight decrease, it is almost half.

Response to R2C40: With the uncertainty of channel evaporation ($\sim$20 mm/year), the decrease is not large. Furthermore, the decrease from the 500m and 750m resolutions to the 1000m resolution is much smaller than from 250m to 1000m. The authors will clarify that the decrease is slight between the coarser resolutions (e.g. 750m to 1000m).

R2C41: L39-350: the authors claim that the decrease in annual recharge is largely linked to ET, which is mainly from high Ei. However, I cannot see the Ei results in the suggested Figures (Fig.1, Fig.4, and Fig. S5) or even in any figure.

Response to R2C41: The authors thank the reviewer for their comment. The authors will adjust the statement to remove inference to the interception evaporation on these figures.

R2C42: L363: Again, here I could not see Supplementary Material B.

Response to R2C42: As with Response to R2C28, the authors will revise the link to the supplementary material throughout the manuscript.

R2C43 L370: Can the authors explain why transpiration age is longer than soil in layers

1 and 2 and in GW.

Response to R2C43: With respect to the reviewer, the groundwater age is much older than any other flux or storage. The groundwater age is in years, more than 14 times older than the transpiration age. To further clarify this in the manuscript, the authors will add a note in the figure caption that groundwater age is in years.

R2C44: L371: I am a bit confused here. I see that it is longer and not lower. e.g. at 250 model resolution, it is 47 days while at 1000 model resolution it is 28 days.

Response to R2C44: The reviewer is correct, the authors intended to state "older" rather than "lower". The authors will change the wording to the correct description.

R2C45: L381: Remove Figure S5. I do not see the results in Fig. S5. Fig. S5 is for the next sentence.

Response to R2C45: The authors thank the reviewer for their suggestion. The authors will move the reference for Fig. S5 to the next sentence.

R2C46: L384-385: here the authors discuss the GW age, however, I could not see the results. Is it in Figure S5? Do the authors mean GW age as L3? I only see GW storage.

Response to R2C46: The authors apologize for the confusion and will revise Fig. S5 to state GW age rather than L3 age.

R2C47: L393-395: I could not see the results for stream water age of 0.5 and 1.8 years during large events in Table 6.

Response to R2C47: The authors apologize for any confusion. The values of 0.5 and 1.8 years are representative of only the largest peak events, while Table 6 shows average high, medium, and low flow conditions. The authors will clarify in the manuscript that the values are indicative of the largest peak events (not averaged with other higher flows).

R2C48: L402: In my opinion, it is not minor variability.

Response to R2C48: The authors believe that with respect to the model uncertainty, there is little difference in the catchment-wide annual average fluxes are relatively similar between resolutions. There are of course larger spatial differences between resolutions which the authors discuss later in the discussion section. The authors will add the qualifier that differences are minor relative to model uncertainty.

R2C49: L407-409: see my general comment point 4.

Response to R2C49: The authors will revise the manuscript as suggested by general comment 4.

R2C50: L467: it is "are" and not "is"

Response to R2C50: The authors will revise the grammar of the sentence.

R2C51: L482: remove the second is in between coupling and with.

Response to R2C51: The authors will remove the second "is".

R2C52: L486: the authors may revise the sentence into: "...scales, long-term and multiscale data collection are..."

Response to R2C52: The authors will make the reviewers suggested wording change.

R2C53: L489-490: the authors may revise the text in the brackets as (e.g. spatial resolution for isotopes and temporal resolution for sapflow).

Response to R2C53: The authors will make the wording change suggested by the reviewer.

R2C54: L498: in my opinion, I will limit to 500 m maximum due to the highest uncertainty in model results above 500 m (Figure 3, 5).

Response to R2C54: The authors agree that of the four resolutions presented here, the maximum resolution to use is 500 m. However, as model resolutions between 500

m and 750 m were not explored here, the Representative Elementary Area may be between 500 m and 750 m.

R2C55: L500: Figures 6e-h are not the correct figures.

Response to R2C55: The authors will change the figure reference to Fig. 5e-h.

R2C56: L516: the authors may remove the words "have been identified"

Response to R2C56: The authors will remove "have been identified"

R2C57 L526-527: missing "that"...fluxes that are additionally expected to change is not well known.

Response to R2C57: The authors will change the sentence to " ...regulating ecohydrological fluxes that are ..."

R2C58 P26: Figure 3. Swap a, b for discharge first. To increase figure readability, the map can be removed since it is already available in Fig. 1a.

Response to R2C58: The authors kindly refer to the reviewer to Response to R2C34.

R2C59 P28: Figure 5. I could not see the measured SM (dashed line).

Response to R2C59: The authors will revise Figure 5 to make the measured SM more apparent.

R2C60 P31: Table 1. Suggestion: please indicate in the table which one is obtained from ERA5 and which one is from MODIS. e.g. instead of location N/A, why not write ERA5 or MODIS? Also please keep in mind ERA 5 is not categorized as remote sensing data.

Response to R2C60: The authors thank the reviewer for their suggestion. The authors will modify Table 1 to indicate which dataset originates from MODIS and ERA5 and will modify the caption to better reflect reanalysis and remote sensing data.

R2C61: P32: missing Table 2 borderline.

Response to R2C61: The authors will add borderlines to Table 2.

R2C62: P34: Table 4. What is negative Loglikelihood? It is not explained in the text.

Response to R2C62: To ease with interpretation and reduce complexity, the authors will remove the loglikelihood values from Table 4.

R2C63: P35: Table 5. Please mention that Es, Ei, and Tr were partitioned from ET.

Response to R2C63: Within EcH2O, Es, Ei, and Tr are estimated first with ET the sum of the components rather than partitioning ET into each. The authors recognize that this was not clear within the manuscript (L330), and will clarify the estimation of ET components.

---

## Author Response (AR1)

**Response to Reviewers**

Dear Dr Wang

On behalf of the authors, we would like to thank the editor for the opportunity to revise the manuscript. Through the revision, the authors have aimed to address the comments provided by the reviewers. These revisions include clarifying the objectives through the manuscript (introduction and conclusion), expanding the calibration to include more detailed information on the multi-criteria method, and elaborating on the discussion of the modelled ET components. The authors believe that this revision has aided in making the manuscript clearer and has improved the readability.

Sincerely,

Aaron Smith

**Reviewer 1**

*General Comments*

The paper presents substantial new results on the effects of model resolution on ecohydrological flux simulation using supplementary isotope data. The overall quality of the paper is very good, but some key aspects of the methodology are unclear in the main manuscript, and are only outlined in the supplementary material. Other than a few details, the paper is clear and well supported. I see no reason not to accept the manuscript for publication in HESS, subject to a few minor corrections.

*Response to General Comments*
The authors thank the reviewer for their comments. The authors have clarified the methodology presented in the manuscript to reduce dependencies on the supplementary material.

*Specific Comments*

**R1C1:** 74-80: The research questions are clear enough but three fairly complex questions seems excessive for a single research paper. Having read through the manuscript, all the questions are addressed to some degree, but they are an unfocused introduction to the aim and scope of the presented research.

**Response to R1C1:** The authors thank the reviewer for their suggestion. The authors have revised the objectives to be more specific to the presented research. The revision is more specific to objectives 2 and 3. L75-79.

**R1C2:** 86: Describing a 66 km$^2$ catchment as 'mesoscale' is dubious; it falls short even of your previous definition of 100km$^2$. Let the area speak for itself, as was done in the abstract.

**Response to R1C2:** The authors intended for the term 'mesoscale' to be a general descriptor to differentiate the site as being larger than a small experimental catchment (typically <10km$^2$ and often <1km$^2$). The authors have clarified this in the manuscript "…spatial scales through application to a mesoscale (i.e. >10 km2) mixed land-use catchment". L67.

**R1C3:** 210: Is 'climate zone' the best word choice? It doesn't necessarily bring weather stations and Thiessen polygons to mind.

**Response to R1C3:** The authors have revised the wording to "…forcing data were included as representative polygon areas from five local climate stations". L211-213.

**R1C4:** 233: It is stated that NRMSE is used, but not what is used to normalize the error. The justification for using the NRMSE is also insufficient to explain why the NSE was rejected (the NSE is fundamentally a normalized squared error) for the isotope simulations. Either a better justification in text or a reference is needed.

**Response to R1C4:** The authors apologize for the error in the manuscript. The RMSE and NRMSE were exclusively used for the sensitivity analysis. Calibration efficiency was measured by NSE and normalized mean absolute error (NMAE). The authors agree that there is limited usefulness of NRMSE if NSE is used. The NMAE was used because the authors did not want to over-emphasis peak values when data collection had inconsistent time-steps and the isotope response was quite subdued. The authors have revised the section to indicate that NMAE was used for calibration, with NRMSE used for the sensitivity analysis (Section 3.5.1).

**R1C5:** 258-260: The multi-criteria calibration methodology is a key part of this research, but it would not be possible to replicate the method with the description provided here. Too many significant details on the method have been relegated to the supplement.

**Response to R1C5:** The authors thank the reviewer for their suggestion. To keep the manuscript streamlined, the authors have moved the most significant details of the multi-criteria calibration from the supplementary material to the manuscript (L264-268). However, we are mindful of maintaining a balance between detail and excessive manuscript length and using the supplementary material to achieve this.

**R1C6:** 374: Unless there is some kind of character limit on the manuscript, the readability might be improved by using actual process terms rather than contractions. Tr and Re were defined but in a different section.

**Response to R1C6:** To aid with readability and maintain consistency, the authors have removed the abbreviations for transpiration and recharge (Tr and Re) throughout the manuscript, and replaced with the full term.

**R1C7:** 66: Squared kilometres is not properly superscripted. This occurs irregularly throughout the manuscript.

**Response to R1C7:** The authors have checked and corrected errors in sub- and superscripts.

**R1C8:** 81: Likely grammatical error, unsure what the subject of 'sought' is.

**Response to R1C8:** The authors have revised this sentence to "The evaluation of these questions across different spatial model resolutions is aimed at providing a more robust understanding of the spatial boundaries of the ecohydrological exchange, partitioning, and uncertainty in models". L80-81.

**R1C9:** 207: Is there too many 'of' in the length description?

**Response to R1C9:** Corrected. L210.

**R1C10:** 236: Supplementary material referenced by letter, but the supplement is numbered. Not the only location this occurs.

**Response to R1C10:** The supplementary material lettering has been updated to numbers to be consistent with the appended supplementary material.

**R1C11:** Figure 1: There is no scale on the main map in (a)

**Response to R1C11:** The authors have added a scale to the main map of Figure 1a

**Reviewer 2**

*General Comments*

This paper evaluates the performance of an ecohydrological model equipped with water isotope module (EcH2O-iso)in simulating soil water, evapotranspiration, recharge or runoff, groundwater, and water ages across different spatial scales (250, 500, 750, and 1000m). Moreover, the study addresses three main research questions that will be revealed using the model, applied in an intensively monitored catchment in northeast Germany. They conclude that the model with a coarser resolution (1000m) was unable to replicate the observed streamflow and distributed isotope dynamics and simulates higher evapotranspiration, lower relative transpiration, increased overland flow, and slower groundwater movement. In addition, they also conclude that water isotopes provide effective calibration constraints for larger resolution model and help understand the influence of grid-resolution on the simulation of vegetation-soil interactions.

Assessment:

The topic of the paper fits well in HESS and is of great interest for the HESS readers. The findings are also interesting for readers working on the ecohydrological model development, land-surface fluxes quantification, and the use of water isotopes in hydrological applications. The paper at the present state still needs some revisions. The authors need to elaborate more on the method section and the conclusion does not summarize all three research questions brought up in this paper. I provide my general and detailed comments below and would ask the authors to take these comments into consideration as they revise the paper.

1. *In my opinion, the title is not mirroring the aim of this paper. The title is: upscaling land-use effects on water partitioning and water ages using tracer-aided ecohydrological model (L1) while the aim of this paper is to*

*explore the changes in the skill of an ecohydrological model in capturing flux, storage, and mixing dynamics across spatial scales (L65). When I read the paper thoughtfully, the paper discusses the performance of the model to simulate the water fluxes and ages across different spatial scales. Therefore, I suggest to change the title.*

2. *The study tries to answer three research questions (L74-80) and they are explained in the discussion section.*

*For the conclusion section, however, a summary of the answers for these research questions is not clearly presented. I could not find a conclusion for research question two or maybe I miss it.*

3. *The EcH2O-iso model consists of three main parts, which are the energy balance model, the water balance model, and the water isotope module. I understand that the idea of the paper is not to discuss model development in a detailed manner as it was discussed in the previous studies (e.g., Maneta and Silverman,*

*2013 and Kuppel et al., 2018). However, there are not so many hydrological models equipped with water isotope module compared to the climate model. I expected that the authors would provide more information about how isotopic mixing and fractionation are performed in the model. Please elaborate more about the mixing and fractionation in all water components, such as in precipitation, transpiration, soil, storage, and GW. Also what methods do the authors use to calculate isotopic fractionation in different fluxes such e.g. the*

*Craig-Gordon method, Keeling plot, or a steady-state method introduced by Dongmann et al. (1974) to calculate the isotopic composition of leaf water? How to define the isotopic composition of river water since it is mixing between groundwater, precipitation, and river from the upstream. When river water flowing in the channel, does it undergo evaporation fractionation, or neglected?*

4. *In the discussion section, the authors found that for forests, the evapotranspiration is higher, the recharge is*

*lower, and the transpiration fraction is lower compared to croplands (L407-409). They argue that the lower Tr ratio is due to the interception. Could the authors elaborate more on why and what are the possible reasons? Also please provide all the evapotranspiration fractions here (soil evaporation, transpiration, and interception for crops and forest). I found in many studies that taller plants transpire more water than shorter plants (e.g., Oaks vs. wheat, Xu et al., 2008; Zhang et al., 2011, cotton field vs. corn and soybean,*

*Kool et al., 2014). For some field measurements, the transpiration fraction for forests is 65-76% and 60%*
       *for grass (Choudhury and DiGirolamo, 1998; Blyth and Harding, 2011).*

**Response to General Comments**

The authors thank the reviewer for their comments on this manuscript, which have aided in adding clarity in
presenting model results and objectives.

1.   *The authors have revised the title of the manuscript to "Quantifying the effects of land-use and model scale on water partitioning and water ages using tracer-aided ecohydrological models"*
2.   *As suggested by the reviewer, the authors have revised the conclusion to clearly state the summary of the*
*second research goal. L547-549.*
3.   *The authors thank the reviewer for their suggestion to further elaborate on the isotope and water age mixing section. The authors recognize that isotope mixing modules are not present in many models. However, the authors believe that repeated explanation of the same mixing process for each storage would be quite repetitive. To better clarify the mixing approaches taken within this modelling, the authors have clarified that*
*mixing is conducted with amount weighted average of inputs and storage, and how transpiration mixing is conducted (update to section 3.3). The authors have added a sentence to indicate that the Craig-Gordon model is used to estimate the fractionation in the soils and in the open water (L197). The manuscript currently states that open water channel evaporation is subject to fractionation (Previous manuscript L200, revision L204).*
4.   *There are numerous factors that, in this catchment, result in a somewhat unexpected ratio of transpiration to ET in the forested areas. Firstly, it should be noted that the values presented in the manuscript are annual average fractions, which are moderately skewed toward lower transpiration to ET ratios due to winter interception and soil evaporation. This will be particularly marked in the conifer forests where the LAI remains constant year-round facilitating higher interception storage and evaporation. On this point, the*
*authors have added clarification in the discussion on the seasonality of the transpiration to ET ratio influence in the conifer forests (L419-423). To further clarify, the authors have included a time series of the average evapotranspiration components for the dominant vegetation types in the supplementary material (Fig S6) to show this variability. Secondly, while the transpiration to ET ratio is lower in the conifer forests, the dominant soil type in the forests is also different from the croplands, being less water retentive. This soil*
*type limits the water available for the trees, introducing water stress and reducing transpiration further from potential (not energy limited). The observed transpiration to ET ratio in the study is similar to those given by Choudhury and DiGirolamo, 1998 and Blyth and Harding, 2011. To further clarify the reason for the low transpiration due to dry soil conditions, the authors have added to the discussion clarifying the water-limited conditions (L423-426).*

**Specific Comments**

**R2C1:** L1: for the abstract, please see the general comment.

**Response to R2C1:** The author refers the reviewer to the response to the general comments.

**R2C2:** L15: is it not better if the authors write "different scales" instead of only "scale"?

**Response to R2C2:** The authors have revised the wording to "…and the effects of different scales on the skill of ecohydrological… " L16

**R2C3:** L65: here the readers can see the aim of the study, which differs from the title (upscaling land-use effects instead of evaluating the model skill in simulating different water fluxes across different spatial scales).

**Response to R2C3:** As the reviewer suggested in the general comments, the authors have revised the title to better fit the overall aims of the manuscript.

**R2C4:** L76-77: the second question is not clearly answered in the conclusion.

**Response to R2C4:** The authors have added the statement "Identification of sensitive parameters was similar across scales; however, a notable decrease in the degree of sensitivity, coupled with an increase in all model output uncertainty, occurred with coarser model resolutions." to the conclusions to directly address the primary findings of the second research question. L547-549.

**R2C5:** L86-87: the authors may rephrase the sentence into: "The 66 km2 Demnitzer Millcreek Catchment (DMC) located 55 km east of Berlin (52_23'N, 14_14'E) is a mesoscale catchment that receives 575 mm of precipitation annually"

**Response to R2C5:** The authors have revised the wording of the sentence to "The 66 km$^2$ Demnitzer Millcreek Catchment (DMC), is a catchment 55 km east of Berlin (52$^o$23'N, 14$^o$15'E), that receives 575 mm of precipitation annually." L85-86

**R2C6:** L87: the authors may change the words: "(372 to 776 mm/year)" into "from 372 to 776 mm/year".

**Response to R2C6:** The authors have changed the statement to " Cumulative annual precipitation varies from 372 to 776 mm/year; with summer usually…". L86.

**R2C7:** L88: I think the general term is "convective" storms and not convectional storms.

**Response to R2C7:** The authors have changed the term in the manuscript. L87.

**R2C8:** L94: missing comma …unavailable, these were …

**Response to R2C8:** The authors have revised the grammar. L92.

**R2C9:** L95: first, ERA 5 is not a remote sensing dataset. It is re-analysis product that is fussed with all types of observations (not only remote sensing). Second, it is more common and general to use the word "sensing" instead of "sensed". I cannot provide a good argument about it but you may look at:

https://www.researchgate.net/post/Remotesensing-data-or-images-vs-remotely-sensed-data-or-images-Are-they-both-OK

**Response to R2C9:** The authors thank the reviewer for their suggestion. The authors have changed the description of the ERA5 dataset to re-analysis data "…, these were derived from re-analysis data, ERA5…" L93. Throughout the manuscript, the authors have changed instances of "sensed" to "sensing".

**R2C10:** L131: the authors may revise the text into: "…during periods of streamflow measurements". For the next sentence, the authors may revise the sentence to avoid the use of the words "periods of streamflow" two times.

**Response to R2C10:** The authors have removed "during periods of streamflow" for both sentences, and have added a sentence stating "Isotopic samples of stream water were only taken when streams were flowing and not during standing water.". L129-130.

**R2C11:** L132: the authors may revise the text into: "Evaporation was prevented by applying a layer of

**Response to R2C11:** The authors have revised the sentence to "Evaporation was prevented by applying a layer…" L131.

**R2C12:** L134: missing "were" analysed. What is direct-equilibrium method?

**Response to R2C12:** The authors have revised the wording of the sentence to include "were" L132. The authors feel that the full description of the direct-equilibrium method is not necessary for interpreting the manuscript, and have made it more explicit that the full analysis is described in Kleine et al., 2020 for data at these sites. L132.

**R2C13:** L135: the authors may replace the word "with" with "using"

**Response to R2C13:** The authors have modified the sentence to "…were analysed in the IGB laboratory using a Picarro L-2130i cavity ring…". L133.

**R2C14:** L141: what is "cf"?

**Response to R2C14:** With respect, cf is the APA Latin abbreviation of conferatur, meaning: comparable to/to make a comparison.

**R2C15:** L145: what kind of components? It is an unclear word.

**Response to R2C15:** The authors have modified the sentence to "…integrates modules for soil and vegetation to simulate…". L143.

**R2C16:** L146: I am just wondering if the model was designed to be forced with RCM only or can it be forced with gridded in situ observations? This study also used in situ observations and not RCM.

**Response to R2C16:** The authors have revised the sentence to "The model is designed to be forced with inputs either from local climate stations or from regional climate models" to clarify that RCMs are not the only forcing data that may be used. L144-145.

**R2C17:** L149: I suggest to replace ";" with full stop "."

**Response to R2C17:** Corrected. L148.

**R2C18:** L172: Could the authors indicate layers 1 to 3 in Fig. S1? It will help the readers to discriminate layers.

**Response to R2C18:** The addition of layers 1, 2 and 3 would be relatively arbitrary (e.g. water table may vary between layers 1 and 3), and would further complication of the plot due to additional flux arrows to and from each soil layer.

**R2C19:** L197: missing space.
**Response to R2C19:** Corrected. L198.

**R2C20:** L200: please provide a reference for the n values.
**Response to R2C20:** The authors have modified the statement to include the references to the aforementioned Mathieu and Bariac (1996) and Braud et al (2005) papers, ". The kinetic fractionation factor ($n$) is corrected using soil saturation to adjust the n value (liquid-vapour turbulence) between $n = 1$ (dry soil) and $n = 0.5$ (fully saturated soils) (Mathieu and Bariac, 1996; Braud et al., 2005)". L201-202.

**R2C21:** L210: they are locations and not climate zones. Also please provide the names for these five locations in the main text.
**Response to R2C21:** The authors have revised the text to "To reduce the effect of the spatial resolution of climate model forcing data on model results (e.g. Liang et al., 2004), forcing data were included as representative polygon areas from five local climate stations (Table1)". L211-213. As the names of the stations do not have meaning to those outside of the area, the authors believe that the inclusion of the station names is not necessary.

**R2C22:** L221: How about streamflow isotopic composition?
**Response to R2C22:** The authors have revised the statement to indicate that stream water was initialized using measured stream isotopes "Soil, stream, and groundwater isotopic compositions were initialized using soil, stream, and groundwater measurements in 2018 and 2019.". L224-225.

**R2C23:** L224: I am wondering why the use of NSE and NRMSE is inconsistent. Some variables were evaluated using NSE and some using NRMSE.
**Response to R2C23:** The authors apologize for the confusion regarding the efficiency criteria. The RMSE was only used for the sensitivity analysis, while NSE and normalized mean absolute error (NMAE) were used for calibration. MAE was used due to the inconsistent time-series and to reduce over-emphasis of peak values which may be present in time-series with longer periods between samples. The authors have revised the section to indicate that NMAE was used for calibration, with NRMSE used for the sensitivity analysis (Section 3.5.1).

**R2C24:** L247: I am wondering what are the variable constraints do the authors use to calibrate the model? Is it only discharge or the authors consider all variables such as GW, SM.
**Response to R2C24:** The authors have moved some of the calibration descriptions from the supplementary material to the main text to clarify the calibration method. The authors performed multicriteria calibration for the model using discharge, fluxes (ET and transpiration), and isotopes (soil and streamflow) (L253-254). The authors have provided
       an additional reference to Table 1 to directly show the datasets used in calibration and validation.

       **R2C25** L260: Fig. S3 is for validation and not for calibration.
       **Response to R2C25:** As stated in the caption of Fig. S3 in the supplementary material, the figure shows the ranges
of the parameters from calibration.

       **R2C26:** L269-271: I am a bit confused here. How do the authors calibrate the isotopes?
       **Response to R2C26:** The authors apologize for the confusion in the calibration section. Since isotopes were only
       available during the calibration period, the authors split some of the isotope data to use some during validation. The
isotopes (stream, groundwater, and forest layer 1 isotopes) were calibrated with NMAE, and forest layer 2 and
       cropland layer 1 were used for validation. The authors have clarified the data used for multicriteria calibration and
       validation in Table 1 and the text (L253-254).

       **R2C27:** L276: the authors may replace the word "very few parameters" with "a few parameters"
**Response to R2C27:** The authors have changed the sentence to "…that the RMSE of model output is sensitive to a
       few parameters which….". L284.

       **R2C28:** L277: Do I miss supplementary material B? I could not find it.
       **Response to R2C28:** Supplementary Material section 2 was erroneously labelled as Supplementary Material B. The
authors have corrected the reference to Supplementary Material 2. L285.

       **R2C29:** L279: two commas there are not needed.
       **Response to R2C29:** Corrected. L287.

**R2C30:** L284: How about streamflow?
       **Response to R2C30:** The authors have changed the wording of the sentence from "runoff generation" to "discharge.
       L292.

       **R2C31:** L285-287: here the authors mention soil moisture in layers 1 and 2, however, Figure 2c does not distinguish
between layers 1 and 2. Where can I look at the results?
       **Response to R2C31:** Figure 2c shows both layers 1 and 2 (previously shown with headers SWCL1 and SWCL2,
       respectively). To make it more explicit that SWCL1 and SWCL2 indicate the soil moisture in layers 1 and 2, the
       authors have changed the header description and caption to ease interpretation. (Figure 2).

**R2C32:** L295: I suggest to add the word "that" in between processes and were.
       **Response to R2C32:** Corrected. L302.

**R2C33:** L311: I am wondering why the Ei is somehow twice as higher as soil evaporation and half of Tr. If I look at the land cover map (Fig. 1a), I see the land cover, in general, can be divided into half forest and half arable land. The Ei is indeed higher for the forest but it cannot exceed the TR, and EI is very low or even insignificant for arable land. What are the reasons? Do the authors think this is the general problem (underestimation of Tr) found in many models as it was discussed by Sutanto et al., 2014?

**Response to R2C33:** The figures show the annual average components of Tr and Es from ET. Due to the more constant winter precipitation feeding interception, overall transpiration appears lower, particularly in the forests where sandier soils dominate and higher winter groundwater recharge limits the water available for spring/summer forest transpiration. Lower LAI in the arable lands decreases the potential interception evaporation relative to the forest. The sandier soils in the forest additional result in lower soil moisture, resulting in soil moisture limited transpiration flux. For the ratio between interception evaporation and transpiration, the lower atmospheric resistance of interception evaporation results in lower transpiration fractions during extended periods of higher rainfall (i.e. more intercepted water). Additionally, the fractions presented here are consistent with those in Europe, with grasslands/maize (Sutanto et al., 2012; Herbst et al., 1996) showing values of 77-97% transpiration, consistent with the finding in the arable land in our study and a catchment average of 72-77% Tr/ET (Fig. 4). The authors have added this explanation to the discussion. L418-426.

**R2C34:** L316: If the discharge is discussed first, why do not the authors swap the Figure between e and f with a and b? Hence, Figure 3a and B will be for streamflow results.

**Response to R2C34:** The authors have modified Figure 3 to show the discharge first (Fig 3a and b) and have removed the map to aid with readability.

**R2C35:** L324: the authors may revise the sentence into "…flow events that are not present…"

**Response to R2C35:** Corrected. L331.

**R2C36:** L329: It is not the correct sub-title.

**Response to R2C36:** The authors have revised the sub-title to "Effect of model scale on ecohydrological fluxes and storages".

**R2C37:** L330: here the authors only mention Tr. How about the Es and Ei?

**Response to R2C37:** The authors only mention ET and Tr as they were the time series used in calibration. Es and Ei were not used in calibration and therefore are not relevant to include here. As with Response to R2C24, the authors have revised the calibration section to help clarify the data used for calibration.

**R2C38:** L336: I am just wondering why the authors do not use the blue color for positive and red color for negative correlation (inverse colors)? Usually red represents low value and blue represent high value.

**Response to R2C38:** The authors have reversed the colour scheme to show blue for the positive correlation and red for the negative correlation.

**R2C39:** L337: please provide the value for the fraction of Tr.

**Response to R2C39:** The authors have provided a reference to Table 5 and Fig.4 which both show the annual average fraction of Tr (Table 5 with Tr as a fraction of total catchment loss). L343.

**R2C40** L343: from 38 mm/year to 22 mm/year is not a slight decrease, it is almost half.

**Response to R2C40:** With the uncertainty of channel evaporation (~20 mm/year), the decrease is not large.

Furthermore, the decrease from the 500m and 750m resolutions to the 1000m resolution is much smaller than from

250m to 1000m. The authors have clarified that the decrease is slight between the coarser resolutions (e.g. 750m to

1000m). L351.

**R2C40:** L39-350: the authors claim that the decrease in annual recharge is largely linked to ET, which is mainly from high Ei. However, I cannot see the Ei results in the suggested Figures (Fig.1, Fig.4, and Fig. S5) or even in any figure.

**Response to R2C40:** For clarity and to limit the keep the number of plots lower, the authors have changed the statement to "The decrease in annual recharge is largely linked to the higher ET (Fig. 1; Fig. 4; Fig. S5).". L357.

**R2C41:** L363: Again, here I could not see Supplementary Material B.

**Response to R2C41:** As with Response to R2C28, the authors have revised the link to the supplementary material.

L370.

**R2C42** L370: Can the authors explain why transpiration age is longer than soil in layers 1 and 2 and in GW.

**Response to R2C42:** With respect to the reviewer, the groundwater age is much older than any other flux or storage.

The groundwater age is in years, more than 14 times older than the transpiration age. To further clarify this in the manuscript, the authors have added a note in the figure caption that indicates that the groundwater age is in years.

**R2C43:** L371: I am a bit confused here. I see that it is longer and not lower. e.g. at 250 model resolution, it is 47

days while at 1000 model resolution it is 28 days.

**Response to R2C43:** The reviewer is correct, the authors intended to state "older" rather than "lower". This error has been revised. L378.

**R2C44:** L381: Remove Figure S5. I do not see the results in Fig. S5. Fig. S5 is for the next sentence.

**Response to R2C44:** The authors thank the reviewer for their suggestion. The authors have moved the reference for

Fig. S5 to the next sentence. L390.

**R2C45:** L384-385: here the authors discuss the GW age, however, I could not see the results. Is it in Figure S5? Do the authors mean GW age as L3? I only see GW storage.

**Response to R2C45:** The authors apologize for the confusion and have revised Fig. S5 to state GW age rather than L3 age.

**R2C46:** L393-395: I could not see the results for stream water age of 0.5 and 1.8 years during large events in Table 6.

**Response to R2C46:** The values of 0.5 and 1.8 years are representative of only the largest peak events, while Table 6 shows average high, medium, and low flow conditions. The authors have clarified in the manuscript that the values are indicative of the largest peak events (not averaged with other higher flows) "During the largest events, stream water ages dropped most notably in the 750 and 1000 m resolutions (average stream water age of 0.5 years during peak events, $Q_a \gg 1$) reflecting extensive overland flow simulations (Table 6). Stream water ages for the finer grids also decreased during large events (average stream water age of 1.8 years during peak events, $Q_a \gg 1$); however, the change was not quite as large relative to the long-term average stream water age compared to coarse grids.".L399-403.

**R2C47:** L402: In my opinion, it is not minor variability.

**Response to R2C47:** The authors believe that with respect to the model uncertainty, there is little difference in the catchment-wide annual average fluxes are relatively similar between resolutions. There are of course larger spatial differences between resolutions which the authors discuss later in the discussion section. The authors have added the qualifier that differences are minor relative to model uncertainty. L410.

**R2C48:** L407-409: see my general comment point 4.

**Response to R2C48:** The authors refer the reviewer to the response to general comment 4.

**R2C49:** L467: it is "are" and not "is"

**Response to R2C49:** Corrected. L481.

**R2C50:** L482: remove the second is in between coupling and with.

**Response to R2C50:** Corrected. L496.

**R2C51:** L486: the authors may revise the sentence into: "…scales, long-term and multiscale data collection are…"

**Response to R2C51:** Corrected. L498.

**R2C52:** L489-490: the authors may revise the text in the brackets as (e.g. spatial resolution for isotopes and temporal resolution for sapflow).

**Response to R2C52:** Corrected. L502.

**R2C53:** L498: in my opinion, I will limit to 500 m maximum due to the highest uncertainty in model results above

500 m (Figure 3, 5).

**Response to R2C53:** The authors agree that of the four resolutions presented here, the maximum resolution to use is 500 m. However, as model resolutions between 500 m and 750 m were not explored here, the Representative Elementary Area may be between 500 m and 750 m. The authors have added a statement that indicates that the 500m resolution is the maximum representative area presented in this study. L512.

**R2C54:** L500: Figures 6e-h are not the correct figures.

**Response to R2C54:** Corrected. L513.

**R2C55:** L516: the authors may remove the words "have been identified"

**Response to R2C55:** Removed.

**R2C56** L526-527: missing "that"…fluxes that are additionally expected to change is not well known.

**Response to R2C56:** The authors have changed the sentence to " …regulating ecohydrological fluxes that are …". L538.

**R2C57** P26: Figure 3. Swap a, b for discharge first. To increase figure readability, the map can be removed since it is already available in Fig. 1a.

**Response to R2C57:** The authors kindly refer to the reviewer to Response to R2C34.

**R2C58** P28: Figure 5. I could not see the measured SM (dashed line).

**Response to R2C58:** The authors have revised Figure 5 to make the measured SM more apparent.

**R2C59** P31: Table 1. Suggestion: please indicate in the table which one is obtained from ERA5 and which one is from MODIS. e.g. instead of location N/A, why not write ERA5 or MODIS? Also please keep in mind ERA 5 is not categorized as remote sensing data.

**Response to R2C59:** The authors thank the reviewer for their suggestion. The authors have modified Table 1 to indicate which dataset originates from MODIS and ERA5. Additionally, the authors have changed the caption to better reflect reanalysis and remote sensing data.

**R2C60** P32: missing Table 2 borderline.

**Response to R2C60:** Corrected.

**R2C61** P34: Table 4. What is negative Loglikelihood? It is not explained in the text.

**Response to R2C61:** To ease with interpretation and reduce complexity, the authors have removed the loglikelihood values from Table 4.

**R2C62** P35: Table 5. Please mention that Es, Ei, and Tr were partitioned from ET.

**Response to R2C62:** Within EcH2O, Es, Ei, and Tr are estimated first with ET the sum of the components rather than partitioning ET into each. The authors recognize that this was not clear within the manuscript (L330), and have clarified the estimation of ET components.

---

## Author Response (AR2)

**General comment Reviewer 1**

I thank the authors for their work in responding to my comments. I find the paper to be acceptable for publication. I suggest a few, mostly typographical corrections and suggestions below.

**Response to Reviewer 1**

The authors thank the reviewer for their review of the revised manuscript. The authors have addressed all of the reviewer's comments within the manuscript.

**General comments Reviewer 2**

The paper presents substantial new results on the effects of model resolution on ecohydrolgical flux simulation using supplementary isotope data. The overall quality of the paper is very good, and the author's edits have improved the description of the methodology and the discussion of the results. While no further revisions are necessary prior to publication, the authors may wish to note a couple of minor points arising from the previous revisions:

237: NMAE was chosen for 'reducing overemphasis of peak values', which, strictly speaking, implies that peak values are still being overemphasized. Do the authors actual find that NMAE overemphasizes peak values, or should this be corrected?

248: Following revisions, the manuscript states the model was calibrated using a combination of NSE and NMAE, but the sensitivity was assessed using the RMSE. The metric used in the sensitivity analysis can influence the both the relative sensitivity, and in some cases the ranking, of a parameter. Was the potential divergence between the RMSE and the NMAE sensitivity considered?

**Response to Reviewer 2**

The authors thank the reviewer for their comments. As the reviewer has noted, the NMAE does not overemphasize the peak values so the statement in the manuscript has been revised accordingly. For the sensitivity analysis, the authors did conduct sensitivity analysis with multiple different efficiency criteria. As the reviewer suggested, the efficiency criteria considered did in some cases change the ranking of the parameters; however, the most sensitive parameters (for example the 10 most sensitive parameters for discharge) did not change. Furthermore, the output used in the sensitivity analysis was the same (except for Layer 2 soil moisture) as the output calibrated with NSE. Therefore, the authors chose to show the RMSE sensitivity as it best reflects the sensitivity of the calibration.